# DEALing with Image Reconstruction: Deep Attentive Least Squares

**Mehrsa Pourya** [1]  **Erich Kobler** [2]  **Michael Unser** [1]  **Sebastian Neumayer** [3]

## Abstract

State-of-the-art image reconstruction often relies on complex, abundantly parameterized deep architectures. We propose an alternative: a data-driven reconstruction method inspired by the classic Tikhonov regularization. Our approach iteratively refines intermediate reconstructions by solving a sequence of quadratic problems. These updates have two key components: (i) learned filters to extract salient image features; and (ii) an attention mechanism that locally adjusts the penalty of the filter responses. Our method matches leading plug-and-play and learned regularizer approaches in performance while offering interpretability, robustness, and convergent behavior. In effect, we bridge traditional regularization and deep learning with a principled reconstruction approach.
**Code**: https://github.com/mehrsapo/DEAL.

## 1. Introduction

Image reconstruction plays a fundamental role in computational imaging and computer vision (McCann & Unser, 2019; Zeng, 2001). The task is to recover an unknown image of interest $\mathbf{x} \in \mathbb{R}^d$ from noisy measurements $\mathbf{y} \in \mathbb{R}^M$. Their relation is modeled as $\mathbf{y} = \mathbf{Hx}$, where the forward operator $\mathbf{H} \in \mathbb{R}^{M \times d}$ encodes the acquisition process. When $\mathbf{H}$ is ill-conditioned, one can resort to the regularized reconstruction

$$\hat{\mathbf{x}} \in \underset{\mathbf{x} \in \mathbb{R}^d}{\arg\min} \left( \|\mathbf{Hx} - \mathbf{y}\|_2^2 + \lambda \mathcal{R}(\mathbf{x}) \right). \quad (1)$$

The data fidelity $\|\mathbf{Hx} - \mathbf{y}\|_2^2$ controls the consistency of the reconstruction with the measurements. The regularizer $\mathcal{R} \colon \mathbb{R}^d \to \mathbb{R}_{\geq 0}$ encodes prior information about the solution and is also intended to make the problem well-posed.

[1]Biomedical Imaging group, EPFL Lausanne, Switzerland
[2]Institute for Machine Learning, JKU Linz, Austria [3]Faculty of Mathematics, TU Chemnitz, Germany. Correspondence to: Mehrsa Pourya <mehrsa.pourya@epfl.ch>.

*Proceedings of the $42^{nd}$ International Conference on Machine Learning*, Vancouver, Canada. PMLR 267, 2025. Copyright 2025 by the author(s).

Both terms are balanced by the hyperparameter $\lambda \in \mathbb{R}_{\geq 0}$. Throughout this paper, $\mathbf{x}$ is the vectorized version of a (color or grayscale) image of size $(N_{\mathrm{in}} \times H \times W)$.

From classic signal processing to the advent of deep learning, substantial research efforts have focused on designing regularizers $\mathcal{R}$. In the context of data-driven methods, two primary approaches have emerged: (i) the explicit modeling of $\mathcal{R}$; and (ii) the modeling of operators associated with $\mathcal{R}$, such as its proximal operator, which is required in plug-and-play (PnP) reconstruction (Venkatakrishnan et al., 2013). Following the explicit approach, the starting point for this work is the fields-of-experts (Roth & Black, 2009), which reads

$$\mathcal{R}_{\mathbf{m}} \colon \mathbf{x} \mapsto \sum_{c=1}^{N_C} \langle \mathbf{m}_c, \psi_c(\mathbf{W}_c \mathbf{x}) \rangle. \quad (2)$$

Here, each $\mathbf{W}_c \in \mathbb{R}^{HW \times d}$ convolves $\mathbf{x}$ with the filter template $w_c \in \mathbb{R}^{N_{\mathrm{in}} \times K \times K}$ where $K$ denotes the spatial kernel size. Then, the nonnegative potentials $\psi_c \in \mathcal{C}(\mathbb{R})$ are applied entry-wise to the $\mathbf{W}_c \mathbf{x}$. Finally, the weights $\mathbf{m}_c \in [\epsilon_{\mathrm{M}}, 1]^{HW}$ with $\epsilon_{\mathrm{M}} > 0$ determine the (spatially varying) contribution of $\psi(\mathbf{W}_c \mathbf{x})$. Every component of (2) can be learned. Most implementations to date use $\mathbf{m}_c = \mathbf{1}$. They differ in the parameterization of $\psi_c$ and $\mathbf{W}_c$ for the learning process (Chen et al., 2014; Goujon et al., 2024; Zach et al., 2024) or include nonlinear feature transforms (Li et al., 2020; Kobler et al., 2022). While the learning of $\mathbf{W}_c$ and $\psi_c$ have been studied extensively, the local weights $\mathbf{m}_c$ have received little attention so far.

The use of spatially varying $\mathbf{m}_c$ in (2) is referred to as anisotropic regularization. For instance, $\mathbf{m}_c = \mathbf{M}(\mathbf{y})$ can be derived from the data $\mathbf{y}$ using heuristics (Chan et al., 2008; Grasmair & Lenzen, 2010) or a neural network (Kofler et al., 2023; Lefkimmiatis & Koshelev, 2023). When $\mathbf{M}$ extracts features from an estimated reconstruction, it is natural to consider an iterative refinement of $\mathbf{m}_c$. Specifically, the reconstruction from (1) can be fed back into $\mathbf{M}$ to obtain an *improved* $\mathbf{m}_c$ for (2). This leads to the attentive-reconstruction process

$$\mathbf{x}_{k+1} \in \underset{\mathbf{x} \in \mathbb{R}^d}{\arg\min} \left( \|\mathbf{Hx} - \mathbf{y}\|_2^2 + \lambda \mathcal{R}_{\mathbf{M}(\mathbf{x}_k)}(\mathbf{x}) \right) \quad (3)$$

studied by Pourya et al. (2024) for $\psi_c = |\cdot|$. We note that their scheme is computationally demanding since each of

their updates requires one to solve a problem that involves a least absolute shrinkage and selection operator (LASSO).

**Contribution**   Our contributions are as follows.

- We simplify the updates (3) by choosing $\psi_c(\cdot) = (\cdot)^2$ for the $\mathcal{R}_{\mathbf{M}(\mathbf{x}_k)}$ from (2). Then, each update amounts to the solution of a linear equation, which we handle efficiently through the conjugate-gradient method.

- On the theoretical side, we establish the uniqueness of each update in (3), the existence of a fixed point, a condition for the convergence of (3), and a stability result for the resulting reconstruction operator.

- We learn the filters $\{\mathbf{W}_c\}_{c=1}^{N_C}$ and the attention mechanism $\mathbf{M}$ for $\mathcal{R}_{\mathbf{M}(\mathbf{x}_k)}$ through denoising in such a way that the iterations (3) converge. Then, given an inverse problem with forward $\mathbf{H}$, we deploy our method by tuning only two scalar hyperparameters: (i) the noise level; and (ii) the regularization strength $\lambda$.

- Our experimental evaluations show results on par with state-of-the-art methods for various inverse problems.

- We empirically demonstrate the convergence and robustness of our method. We also provide visual interpretations of the learned attention mechanism. Figure 7 is particularly striking: Each pixel of the denoised image is a weighted average of the noisy data, with the weights being well-adapted to the image structure. This qualitative aspect goes beyond numerical metrics and opens the door to well-performing and explainable image reconstructions.

## 2. Related Literature

**Learned Regularization**   Classic regularizers for (1) leverage sparsity in various domains, such as image gradients (Rudin et al., 1992) or wavelets (Mallat, 1999). The parametric model (2) introduced by Roth & Black (2009) has since spurred extensive research. Key areas of investigation include learning paradigms (Chen et al., 2014; Effland et al., 2020), parameterization strategies (Zach et al., 2024), and intrinsic properties like convexity (Goujon et al., 2023; 2024). More complex architectures build upon strategies such as autoencoders (Li et al., 2020), algorithm unrolling (Kobler et al., 2022), adversarial training (Lunz et al., 2018; Prost et al., 2021), and energy modeling (Zach et al., 2023).

In parallel, implicit regularization methods were developed. Here, off-the-shelf denoisers have been incorporated into iterative reconstruction algorithms in place of the proximal operators (Venkatakrishnan et al., 2013; Romano et al., 2017; Zhang et al., 2022). Non-expansive or homogeneous

denoisers lead to a convergent scheme, but it is difficult to enforce these conditions in the learning (Reehorst & Schniter, 2018; Hertrich et al., 2021). Recently, weaker conditions have been proposed by Pesquet et al. (2021); Hurault et al. (2022b).

**Spatial Adaptivity**   An overview of spatially adaptive regularizers (2) is given in Pragliola et al. (2023). Meanwhile, Hintermüller et al. (2017); Van Chung et al. (2017); Kofler et al. (2023) focus on the total-variation (TV) regularizer (Rudin et al., 1992) as a special case, using either heuristics or deep learning to compute the $\mathbf{m}_c$. More general instances are considered by Lefkimmiatis & Koshelev (2023); Neumayer et al. (2023); Neumayer & Altekrüger (2025). The first work deploys non-smooth potentials $\psi_c$ and majorization minimization to solve the problem in (1). Similar to (3), this leads to a series of quadratic problems. The latter works deploy differentiable $\psi_c$ and solve (1) with accelerated gradient descent. All approaches have in common that they update the weights $\mathbf{m}_c$ only once. In particular, they do not refine $\mathbf{m}_c$ and the reconstruction iteratively as in (3).

**Iterative Refinement**   Lenzen et al. (2014); Lenzen & Berger (2015) propose to iteratively refine the weights $\mathbf{m}_c$ for TV and update them using some heuristic. For the general model (2) with $\psi_c = |\cdot|$, a refinement based on neural networks was considered by Pourya et al. (2024). Their $\mathbf{M}$ has a simple architecture comparable to ours. Iterative refinements are also found outside of this setting, for example, in Saharia et al. (2023) for superresolution and in Darestani et al. (2024) for magnetic resonance imaging (MRI).

**Nonlocal Laplacians**   Quadratic potentials $\psi_c$ lead to a symmetric positive semi-definite system as the optimality condition for (3). Another approach to such updates is (iterative) filtering with carefully designed graph Laplacians (Pang & Cheung, 2017). Recently, this was incorporated into deep architectures for image denoising (Zeng et al., 2019; Valsesia et al., 2020) and scene flow (Teed & Deng, 2021).

**Attention Mechanisms**   Originating in natural language processing (Vaswani et al., 2017), attention is now widely used in imaging (Dosovitskiy et al., 2020). Two main approaches exist: (i) images are divided into patches and processed through scaled dot-product attention, as in SwinIR and Restormer (Liang et al., 2021; Zamir et al., 2022); and (ii) attention is integrated into the architecture in the form of pointwise multiplications, as in attention-guided convolutional neural networks (CNN) (Tian et al., 2020). While patch-based methods effectively capture long-range dependencies, they introduce a significant computational cost due to the need to determine inner products across all patches. In contrast, pointwise attention is more economical.

## 3. Methodology

We specify the regularizer (2) using quadratic potentials as

$$\mathcal{R}_{\mathbf{m}}(\mathbf{x}) = \sum_{c=1}^{N_\mathrm{C}} \langle \mathbf{m}_c^2, (\mathbf{W}_c\mathbf{x})^2 \rangle = \|\mathbf{M}\mathbf{W}\mathbf{x}\|_2^2, \quad (4)$$

with the shorthands $\mathbf{W} = [\mathbf{W}_1^\top \ \cdots \ \mathbf{W}_{N_\mathrm{C}}^\top]^\top$ and $\mathbf{M} = \mathbf{Diag}(\mathbf{m}_1, \ldots, \mathbf{m}_{N_\mathrm{C}})$, where $\mathbf{Diag}$ returns a diagonal matrix whose diagonal entries are the argument vectors. This leads to a quadratic reconstruction problem with solutions

$$\hat{\mathbf{x}} \in \underset{\mathbf{x} \in \mathbb{R}^d}{\arg\min} \left( \|\mathbf{H}\mathbf{x} - \mathbf{y}\|_2^2 + \lambda \|\mathbf{M}\mathbf{W}\mathbf{x}\|_2^2 \right). \quad (5)$$

The question then arises as how to properly choose $\mathbf{M}$ and $\mathbf{W}$. Regarding a filter-based interpretation, $\mathbf{W}$ should be independent of $\boldsymbol{y}$, while $\mathbf{M}$ modulates the response $\mathbf{W}\mathbf{x}$ at each location. Ideally, this modulation ought to depend on the structure of the (unknown) $\hat{\mathbf{x}}$. With abuse of notation, we introduce $\mathbf{M} \colon \mathbb{R}^d \to [\epsilon_\mathrm{M}, 1]^{N_C HW}$, which leads to our attentive refinement scheme

$$\mathbf{x}_{k+1} \in \mathcal{T}(\mathbf{x}_k, \mathbf{y}) \quad \text{with} \quad \mathbf{x}_0 = \mathbf{0} \in \mathbb{R}^d, \quad (6)$$

$$\mathcal{T}(\mathbf{z}, \mathbf{y}) = \underset{\mathbf{x} \in \mathbb{R}^d}{\arg\min} \left( \|\mathbf{H}\mathbf{x} - \mathbf{y}\|_2^2 + \lambda \|\mathbf{M}(\mathbf{z})\mathbf{W}\mathbf{x}\|_2^2 \right). \quad (7)$$

The update (6) can be interpreted as an infinite-depth neural network. Hence, we name it deep attentive least squares (DEAL) for image reconstruction. If $\mathbf{x}_k \to \hat{\mathbf{x}}$, we get that $\hat{\mathbf{x}} \in \mathcal{T}(\hat{\mathbf{x}}, \mathbf{y})$, namely, that $\hat{\mathbf{x}}$ is a fixed-point of the operator (7). We restrict $k < K_\mathrm{out}$ and terminate the iterations (6) when $\|\mathbf{x}_{k+1} - \mathbf{x}_k\|_2 / \|\mathbf{x}_k\|_2 \leq \epsilon_\mathrm{out}$ with $\epsilon_\mathrm{out} > 0$.

### 3.1. Architecture

Next, we specify how the DEAL iterates (6) can be cast as a deep neural-network structure. Figure 1 showcase the interplay of its essential building blocks (reconstruction and mask generation), which repeatedly exchange information.

#### 3.1.1. RECONSTRUCTION BLOCK

At the heart of DEAL, the reconstruction block solves the spatially adapted optimization problem in (7) for given attentive weights $\mathbf{M}(\mathbf{x}_k)$ (see Section 3.1.2). The optimality condition for the problem in (7) is given by the linear system

$$\mathbf{A}_k \mathbf{x}_{k+1} = \mathbf{b}, \quad (8)$$

with $\mathbf{A}_k = \mathbf{H}^\top \mathbf{H} + \lambda \mathbf{W}^\top \mathbf{M}(\mathbf{x}_k)^2 \mathbf{W}$ and $\mathbf{b} = \mathbf{H}^\top \mathbf{y}$. The multi-convolution block $\mathbf{W}$ (see Section 3.1.3) is learnable. The hyperparameter $\lambda \in \mathbb{R}$ needs to be tuned for inverse problems outside the training regime. To avoid scaling ambiguities, we impose $\|\mathbf{W}\|_2 = 1$ by spectral

normalization. The data $\mathbf{y}$ and the forward $\mathbf{H}$ are problem-specific inputs that are not learnable. We solve (8) by the conjugate-gradient (CG) algorithm with $\mathbf{x}_k$ as the initial guess. We use a batched CG with at most $K_\mathrm{in}$ steps, where each sample terminates individually if its residue satisfies that $\|\mathbf{A}_k\mathbf{x}_{k+1} - \mathbf{b}\|_2^2 \leq \epsilon_\mathrm{in}$ for $\epsilon_\mathrm{in} > 0$.

#### 3.1.2. MASK-GENERATION BLOCK

To estimate $\mathbf{M}$ for (5) from local image structures, we use the shallow CNN with learnable nonlinearities

$$\mathbf{M}(\mathbf{x}) = (\boldsymbol{\phi}^\sigma \circ \mathbf{W}_\mathrm{mix}^2 \circ \varphi_2 \circ \mathbf{W}_\mathrm{mix}^1 \circ \varphi_1 \circ \mathbf{W}_\mathrm{mask})(\mathbf{x}) \ (9)$$

(see Figure 1 Right). Here, $\sigma$ denotes the model noise level and needs to be tuned outside the training regime. This choice of architecture is inspired by anisotropic diffusion (Weickert, 1998; Bredies & Lorenz, 2018), where $\mathbf{M}$ is typically computed from the gradients of a smoothed image using pixel-wise nonlinearities.

The first multi-conv layer $\mathbf{W}_\mathrm{mask}$ (see Section 3.1.3) in (9) extracts $N_\mathrm{C}$ spatial features using the same architecture $\mathbf{W}$ does in the reconstruction block. The two subsequent convolution layers $\mathbf{W}_\mathrm{mix}^1$ and $\mathbf{W}_\mathrm{mix}^2$ mix the $N_\mathrm{C}$ feature channels using kernels of size $(3 \times 3)$ without bias. The layers are connected via learnable pointwise nonlinearities $\varphi_1$ and $\varphi_2$, for which we follow Bohra et al. (2020). Specifically, each $\varphi_n$ is parameterized as a linear spline with $N_n$ equally distributed knots on $[0, r]$. On $(r, \infty)$ we extend the splines linearly and enforce symmetry by setting $\varphi(x) = \varphi(-x)$ if $x < 0$. In addition, we constrain them to be increasing for $x > 0$. (The removal of either constraint has not led to significant performance improvement.) To guarantee the numerical stability of (8), the output of $\mathbf{M}$ must remain in $(\epsilon_\mathrm{M}, 1]$. Hence, we process each channel $c \in \{1, \ldots, N_\mathrm{C}\}$ individually as

$$\phi_c^\sigma(x) = \max(\min(\varphi_3(\alpha_c(\sigma)\mathbf{x}), 1), \epsilon_\mathrm{M}), \quad (10)$$

where $\varphi_3$ is a symmetric linear spline as before. In contrast to the former splines, $\varphi_3$ must be non-increasing on $[0, \infty)$. The underlying rationale is that $\mathbf{M}(\mathbf{x})$ should be close to 1 for small filter responses (constant image regions) and close to 0 for strong filter responses (salient edges). Following Goujon et al. (2024), we enable multilevel noise training using the positive scalings

$$\alpha_c(\sigma) = \mathrm{e}^{s_c(\sigma)}/(\sigma + 10^{-5}) \quad (11)$$

with learnable linear splines $\{s_c\}_{c=1}^{N_\mathrm{C}}$. By design of $\mathbf{M}$, the first reconstruction block consists of a non-varying problem.

#### 3.1.3. MULTI-CONV BLOCK

The Multi-Conv block advocated by Goujon et al. (2023) consists of multiple convolution layers without nonlinearities in between. It enables the efficient construction of large

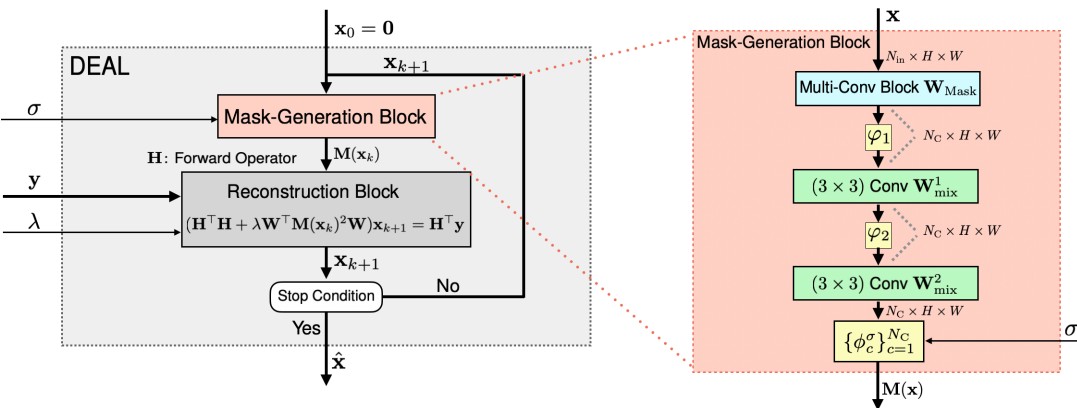

Figure 1: Left: DEAL generates a sequence of reconstructions $\mathbf{x}_{k+1}$ via (6) from the inputs $\mathbf{y}$ and $\mathbf{H}$, initalization $\mathbf{x}_0 = \mathbf{0}$, and hyperparameters $\sigma$ and $\lambda$. When the stop condition is met, it returns $\hat{\mathbf{x}}$. Right: Architecture of the mask-generation block.

receptive fields. We use three convolution layers, all with kernels of size $(9 \times 9)$. Thus, the effective field of view for this block is $(25 \times 25)$. For inputs with $N_{\text{in}}$ channels, the number of output channels are $4N_{\text{in}}$, $8N_{\text{in}}$, and $N_{\text{C}}$ after each of these successive convolution layers. We set the group size and stride to one, and do not use a bias. In all our experiments, we use $N_{\text{C}} = 128$. This block appears in two places: once in $\mathbf{M}(\mathbf{x})$ as $\mathbf{W}_{\text{Mask}}$ and in the reconstruction block as $\mathbf{W}$ (i.e., the convolutions in (4)).

### 3.2. Training

We learn the parameters $\theta$ of DEAL for image denoising with additive white Gaussian noise (AWGN) of standard deviation $\sigma_{\text{n}} \in [0, 50]$. The denoiser $D_{\theta(\sigma_{\text{n}})}^{K_{\text{out}}}(\mathbf{y})$ takes the data $\mathbf{y}$ and $\sigma_{\text{n}}$ as input and then returns the solution of (7) within at most $K_{\text{out}}$ iterations. We have chosen this simple training task for two reasons: (i) our learned model should also work for other inverse problems (often called universality), as demonstrated by Hurault et al. (2022b); Goujon et al. (2024); (ii) our choice leads to simplified computations as $\mathbf{H} = \mathbf{I}$. We provide details on initializations and hyperparameters with a short ablation study in Appendix A.

**Dataset and Loss**  As training set $\mathsf{D} = \{\mathbf{x}^m\}_{m=1}^M$, we use the images proposed in Zhang et al. (2022). The images $\mathbf{x}$ are corrupted by AWGN as $\mathbf{y} = \mathbf{x} + \sigma_{\text{n}}\mathbf{n}$ and fed into DEAL, which leads to the sequence $\{\mathbf{x}_k\}_{k=1}^{K_{\text{out}}} = D_{\theta(\sigma_{\text{n}})}^{K_{\text{out}}}(\mathbf{y})$ of denoised images. To estimate the parameters $\theta$ from the training data, we use the loss

$$\mathcal{L}(\theta) = \left\{ \mathbb{E}_{\substack{\mathbf{x} \sim \mathsf{D} \\ \sigma_{\text{n}} \sim \mathcal{U}([0,50]) \\ \mathbf{n} \sim \mathcal{N}(\mathbf{0}, \mathbf{I})}} \left[ \|\mathbf{x}_{K_{\text{out}}} - \mathbf{x}\|_2^2 \right. \right. \tag{12}$$

$$\left. \left. + \frac{\gamma}{N_{\text{C}}} \|\mathbf{M}(\mathbf{x}_{K_{\text{out}}}) - \mathbf{M}(\mathbf{x}_{K_{\text{out}}-1})\|_2^2 \right] + \gamma \text{TV}^2(\theta) \right\}$$

with $\gamma = 10^{-4}$. This loss consists of three parts: (i) a squared error that enforces that the output matches the clean image; (ii) a squared penalty on the weight changes for the last two updates of (6); and (iii) an accumulated second-order TV regularization of all learnable splines. The latter penalizes changes in the slopes of the splines and thereby promotes simpler splines (Bohra et al., 2020; Ducotterd et al., 2024). The second term in (12) vanishes if the generated weights $\mathbf{M}(\mathbf{x}_k)$ converge. To promote convergence of (6) to a fixed point, we sample $K_{\text{out}}$ uniformly from $[15, 60]$ (Anil et al., 2022).

We minimize the loss (12) using Adam (Kingma & Ba, 2015). At each step of the optimizer, we sample 16 patches of size $(128 \times 128)$ randomly from D. We have two training phases. First, we train the gray and color models for $70\,000$ and $40\,000$ steps, respectively, with an initial learning rate of $5 \times 10^{-4}$ that is reduced to $4 \times 10^{-4}$ by a cosine annealing scheduler. Then, we continue the training of the gray and color model for $10\,000$ and $5000$ steps, respectively, with an initial learning rate of $2 \times 10^{-4}$ that is reduced to $1 \times 10^{-7}$ by annealing. We set $\epsilon_{\text{out}} = \epsilon_{\text{in}} = 1 \times 10^{-4}$ and limit the number of CG steps to $K_{\text{in}} = 50$. To select the best-performing model, we evaluate its performance every 1000 steps and keep the checkpoint with the best validation performance. We use the set3 and set12 datasets to validate the color and grayscale models, respectively.

**Gradient Tracking**  We train DEAL through the deep equilibrium framework (Bai et al., 2019) in the Jacobian free mode. Specifically, we perform at most $(K_{\text{out}} - 1)$ iterations without gradient tracking. Then, after convergence, we perform one additional update (6) with gradient tracking. Here, an efficient backward path for the reconstruction block is crucial, specifically for $\partial_\theta \mathbf{x}_{k+1}$ as defined in (6). Since the backpropagation through the CG algorithm is pro-

hibitively memory-expensive, we propose another approach.

From (8), we have that

$$\mathbf{x}_{k+1} = \mathbf{y} - \lambda \mathbf{L}_k^\top \mathbf{L}_k \mathbf{x}_{k+1}, \tag{13}$$

where $\mathbf{L}_k = \mathbf{M}(\mathbf{x}_k)\mathbf{W}$. The product rule leads to

$$\partial_\theta \mathbf{x}_{k+1} = -\lambda \mathbf{L}_k^\top \mathbf{L}_k \partial_\theta \mathbf{x}_{k+1} - \lambda \partial_\theta (\mathbf{L}_k^\top \mathbf{L}_k) \mathbf{x}_{k+1}. \tag{14}$$

It follows that

$$\mathbf{A}_k \partial_\theta \mathbf{x}_{k+1} = \mathbf{d}_{k+1}, \tag{15}$$

where the matrix $\mathbf{A}_k$ is the one from (8) and only the right-hand side changes to $\mathbf{d}_{k+1} = -\lambda \partial_\theta (\mathbf{L}_k^\top \mathbf{L}_k) \mathbf{x}_{k+1}$. We use auto-differentiation to obtain the gradient estimate $\mathbf{d}_{k+1}$. We then find $\partial_\theta \mathbf{x}_{k+1}$ by solving (15) with CG.

### 3.3. Inference

Once the parameters are learned, we deploy DEAL to a general inverse problem by plugging the corresponding forward $\mathbf{H}$ and its adjoint $\mathbf{H}^\top$ into the reconstruction block. This does not affect the mask-generation block. To adapt to the new task, we only tune two hyperparameters: the model noise level $\sigma$ and the regularization strength $\lambda$ in (6). This requires a small validation set with paired measurements and ground-truth images. Empirically, we observe that the tuning of $\lambda$ is more important than a change in the noise level $\sigma$. A suggested value for $\sigma$ is 15. To ensure that we find a fixed point of (6), we set $K_{\text{in}} = K_{\text{out}} = 1000$, and choose the conservative stop criteria $\epsilon_{\text{in}} = 10^{-8}$ and $\epsilon_{\text{out}} = 10^{-5}$.

## 4. Theoretical Results

All the proofs are provided in Appendix B. Proposition 4.1 guarantees the uniqueness of the updates (6).

**Proposition 4.1.** *If* $\ker(\mathbf{H}) \cap \ker(\mathbf{M}(\mathbf{x}_k)\mathbf{W}) = \{\mathbf{0}\}$, *then* $\mathbf{A}_k$ *is positive-definite and* (8) *has a unique solution. Moreover, if* $\mathbf{M}^2(\mathbf{x}_k) \succeq \epsilon_M \mathrm{Id}$, *then* $\mathbf{A}_k \succeq \mathbf{H}^\top \mathbf{H} + \epsilon_M \mathbf{W}^\top \mathbf{W}$ *and uniqueness holds if* $\ker(\mathbf{H}) \cap \ker(\mathbf{W}) = \{\mathbf{0}\}$.

The next results involve an estimate of the smallest eigenvalue $\lambda_\epsilon = \lambda_{\min}(\mathbf{H}^\top \mathbf{H} + \epsilon_M \mathbf{W}^\top \mathbf{W})$ of $\mathbf{A}_k$.

**Lemma 4.2.** *Let* $\mathbf{x} \in \mathbb{R}^d$. *If* $\ker(\mathbf{H}) \cap \ker(\mathbf{W}) = \{\mathbf{0}\}$ *and* $\mathbf{M}(\mathbf{x})^2 \succeq \epsilon_M \mathrm{Id}$, *then* $\mathcal{T}(\mathbf{x}, \cdot) \colon \mathbb{R}^M \to \mathbb{R}^d$ *is Lipschitz-continuous with constant* $\|\mathbf{H}\|_2 / \lambda_\epsilon$.

Next, we establish the existence of fixed points for $\mathcal{T}(\cdot, \mathbf{y})$.

**Theorem 4.3.** *Assume that* $\ker(\mathbf{H}) \cap \ker(\mathbf{W}) = \{\mathbf{0}\}$ *and* $\mathbf{M}^2(\mathbf{x}) \succeq \epsilon_M \mathrm{Id}$. *Then,* $\mathcal{T}(\cdot, \mathbf{y}) \colon \mathbb{R}^d \to B_r(\mathbf{0})$ *maps into a ball around* $\mathbf{0}$ *with radius* $r = \|\mathbf{H}\mathbf{y}\|_2 / \lambda_\epsilon$. *If* $\mathbf{M}^2 \colon B_r(\mathbf{0}) \to [\epsilon, 1]^{N_{\text{out}} HW}$ *is Lipschitz continuous with constant* $L$, *then* $\mathcal{T}(\cdot, \mathbf{y})$ *admits a fixed point and*

$$\|\mathcal{T}(\mathbf{x}_1, \mathbf{y}) - \mathcal{T}(\mathbf{x}_2, \mathbf{y})\|_2 \leq \frac{L \|\mathbf{H}\mathbf{y}\|_2}{\lambda_\epsilon^2} \|\mathbf{x}_1 - \mathbf{x}_2\|_2. \tag{16}$$

Table 1: Denoising of BSD68 and CBSD68 images.

| | Gray | | | Color | | |
|---|---|---|---|---|---|---|
| $\sigma_\mathrm{n}$ | 5 | 15 | 25 | 5 | 15 | 25 |
| BM3D | 37.54 | 31.13 | 28.61 | 40.19 | 33.52 | 30.71 |
| WCRR | 37.65 | 31.20 | 28.68 | – | – | – |
| SARR | 37.80 | 31.61 | 29.13 | – | – | – |
| SAFI | 37.90 | 31.56 | 29.05 | – | – | – |
| DEAL (Ours) | 37.85 | 31.61 | 29.16 | 40.33 | 33.95 | 31.31 |
| ProxDRUNet | 37.97 | 31.70 | 29.18 | 40.40 | 33.91 | 31.14 |
| DnCNN | – | 31.72 | 29.23 | - | 33.90 | 31.24 |
| DRUNet | **38.09** | **31.94** | **29.48** | **40.59** | **34.30** | **31.69** |

The Lipschitz estimate (16) is very conservative and $\mathcal{T}(\cdot, \mathbf{y})$ appears to often be even a local contraction. If $\mathcal{T}(\cdot, \mathbf{y})$ is contractive for every $\mathbf{y} \in \mathbb{R}^M$, then we get the result of Theorem 4.4.

**Theorem 4.4.** *Assume that* $\ker(\mathbf{H}) \cap \ker(\mathbf{W}) = \{\mathbf{0}\}$ *and* $\mathbf{M}^2(\mathbf{x}) \succeq \epsilon_M \mathrm{Id}$. *If* $\mathcal{T}(\cdot, \mathbf{y}) \colon \mathbb{R}^d \to \mathbb{R}^d$ *is contractive in the sense that if* $\|\mathcal{T}(\mathbf{x}_1, \mathbf{y}) - \mathcal{T}(\mathbf{x}_2, \mathbf{y})\|_2 \leq q\|\mathbf{x}_1 - \mathbf{x}_2\|_2$ *with* $q < 1$ *implies that the iterations* (6) *converge to a unique fixed point* $\hat{\mathbf{x}}$ *and that*

$$\|\mathbf{x}_k - \hat{\mathbf{x}}\|_2 \leq q^{k-1}\|\mathbf{x}_1 - \mathbf{x}_0\|_2. \tag{17}$$

*In particular, we have exponential convergence of* (6)*. Moreover, if* $\hat{\mathbf{x}} = \mathcal{T}(\hat{\mathbf{x}}, \mathbf{y}_1)$ *and* $\hat{\mathbf{z}} = \mathcal{T}(\hat{\mathbf{z}}, \mathbf{y}_2)$*, then it holds that*

$$\|\hat{\mathbf{x}} - \hat{\mathbf{z}}\| \leq \frac{1}{1-q} \frac{\|\mathbf{H}\|_2}{\lambda_\epsilon} \|\mathbf{y}_1 - \mathbf{y}_2\|_2. \tag{18}$$

## 5. Experiments

DEAL is trained with a denoising task. We include the results as baseline benchmark. A key strength of *universal* methods like DEAL is their ability to adapt to different tasks with a simple tuning of the hyperparameters. This is fast and requires only a small amount of task-specific data. To demonstrate the generalization capability, we present results on superresolution and MRI reconstructions, comparing DEAL to selected (mostly universal) baselines. Additional experiments—including grayscale deblurring, hyperparameter sensitivity, and generalization—are given in Appendix C.

### 5.1. Grayscale and Color Denoising

We corrupt ground-truth images by adding white Gaussian noise with standard deviation $\sigma_\mathrm{n} \in \{5, 15, 25\}$. We provide in Table 1 the average peak signal-to-noise ratios (PSNR) achieved by various methods over the images of the BSD68 set and the CBSD68 set, noticing that some approaches are implemented only for grayscale images. First, we include (C)BM3D (Dabov et al., 2007) as a widely regarded

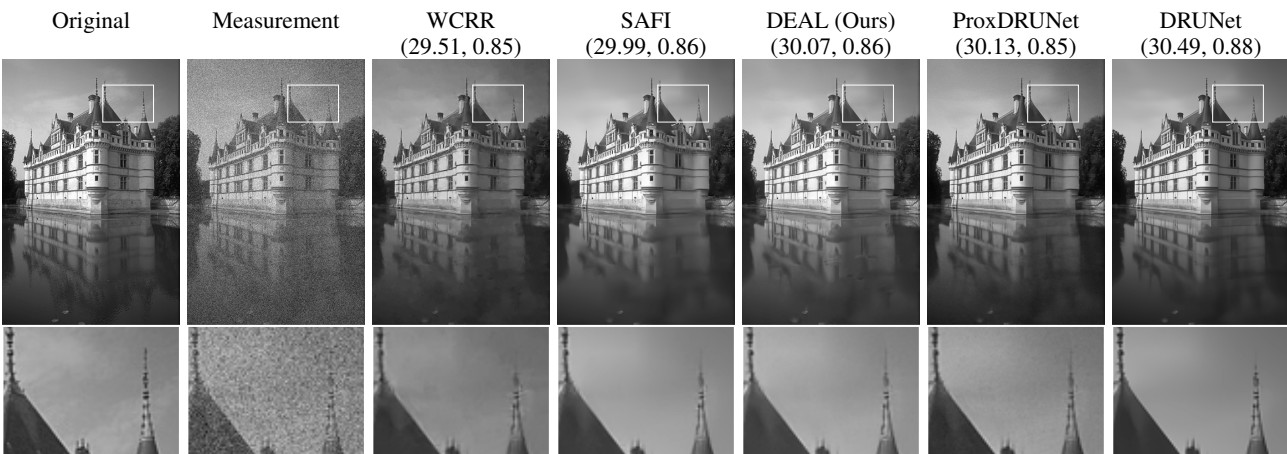

Figure 2: Denoising of the *castle* image for $\sigma_\mathrm{n} = 25$. For each reconstruction (PSNR, SSIM) is provided.

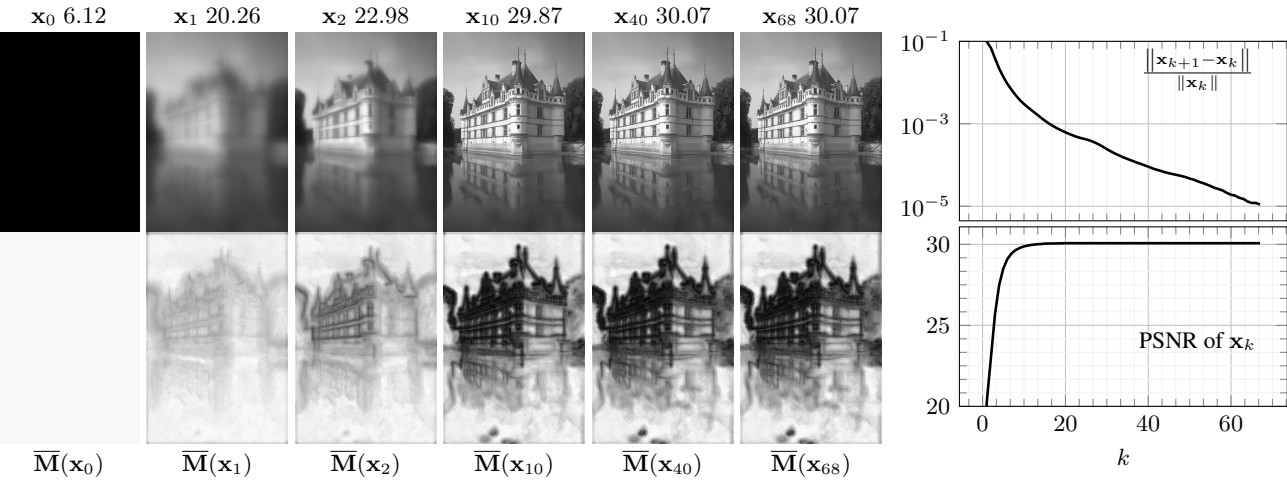

Figure 3: Left: Solution path and channel-wise averages $\overline{\mathbf{M}}$ of the weights for DEAL iterations, exemplified with the *castle* image and $\sigma_\mathrm{n} = 25$. Right: Plot of the residual values and PSNR over the number $k$ of outer iterations.

classic baseline. We also evaluate against WCRR (Goujon et al., 2024), an unadaptive field-of-expert model of the form (2) that employs weakly convex potentials $\psi_c$. We also include its data-adaptive counterpart SARR (Neumayer et al., 2023; Neumayer & Altekrüger, 2025). Regarding the refinement perspective (3), we include SAFI (Pourya et al., 2024), which utilizes $\psi_c = \ell_1$ instead of $\ell_2$. Lastly, we include the deep-learning-based approaches DnCNN (Zhang et al., 2017a), DRUNet (Zhang et al., 2022), and ProxDRUNet (Hurault et al., 2022a). DRUNet is among the state-of-the-art denoisers, and ProxDRUNet is a constrained version of it that is more amenable to PnP reconstructions. DEAL outperforms existing spatially adaptive methods and closes the gap to DRUNet-based approaches while having 30 times fewer parameters. We provide qualitative results in Figure 2, where we also provide the structural similarity-index metric (SSIM). In the magnified part, we can see that DEAL does

better than the DRUNet-based approaches in how it retains structures such as the *tip* of the tower. In Figure 3, we provide the solution path associated to (6), the averages $\overline{\mathbf{M}}$ of the masks $\mathbf{M}(\mathbf{x}_k)$, and two convergence plots. Specifically, the weights $\mathbf{M}$ extract the image structure, leading to lower regularization cost at edge features.

### 5.2. Color Superresolution

Here, the forward $\mathbf{H}$ involves two steps: a blurring of the image through convolution with a known kernel, followed by a downsampling with $s \in \{2, 3\}$ that reduces the number of 2D measurements by a factor of $s^2$. As further degradation, an AGWN of standard deviation $\sigma_\mathrm{n}$ is added to the data. Specifically, we investigate two setups: (i) a bicubic kernel; and (ii) four Gaussian kernels with standard deviations (0.7, 1.2, 1.6, and 2.0) as in (Zhang et al., 2022). We

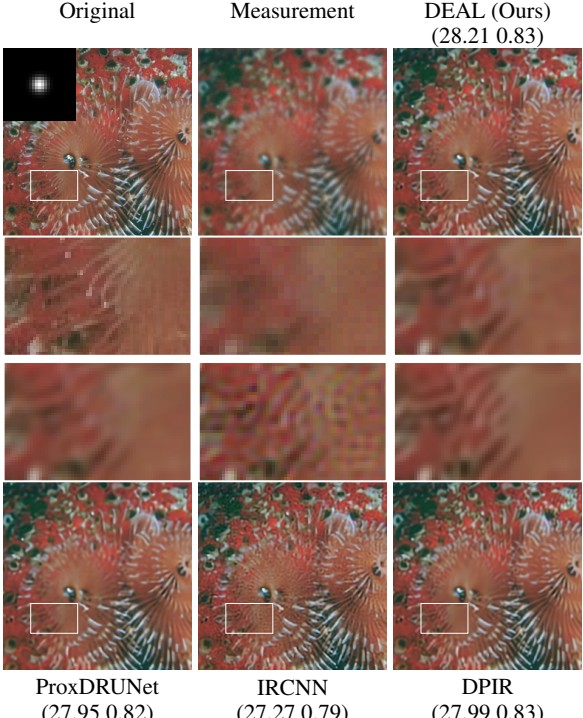

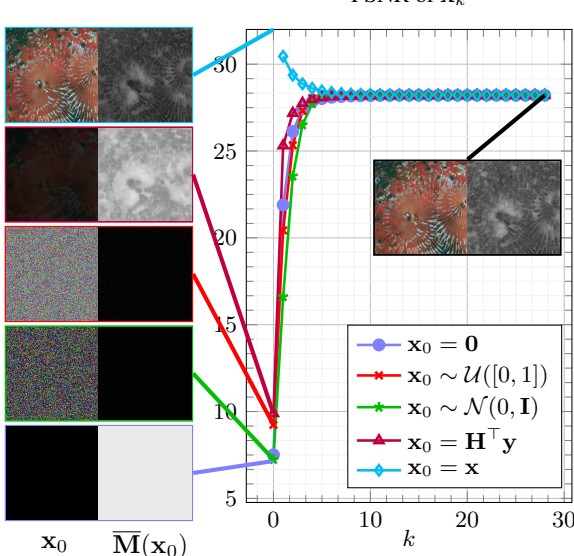

Figure 4: Top: Superresolution task with $s = 2$ and $\sigma_\mathrm{n} = 2.55$. Bottom: Dependence on initialization.

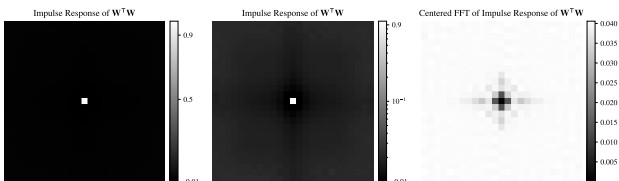

Figure 5: Visualizations of $\mathbf{W}^\top\mathbf{W}$. Only one eigenvalue is zero.

report in Table 2 the average reconstruction PSNR for the center-cropped CBSD68 images and the number of parameters for all models. This evaluation is used by Hurault et al. (2022a), and differs from the one of Zhang et al. (2022). In Figure 4, we provide a visual comparison. Overall, DEAL has a nice complexity-performance tradeoff.

To deploy DEAL, we set $\sigma = 15$ in the architecture trained for denoising and only tune the scalar hyperparameter $\lambda$ in (7) to the noise level $\sigma_\mathrm{n}$, keeping it fixed across the superresolution tasks. This requires only a few validation images. Specifically, we found $\lambda \in \{0.1, 0.28, 2.5, 5.5\}$ for $\sigma_\mathrm{n} \in \{0, 2.55, 7.65, 12.75\}$, respectively. An explicit formula for $\lambda$ is investigated in Table 8 of Appendix C.

We tune the other universal methods based on the recommendations of their papers. DPIR (Zhang et al., 2022) represents the state-of-the-art among PnP methods, but is tied to a fixed number of iterations. ProxDRUnet (Hurault et al., 2022a) addresses this by constraining DRUnet, which empirically ensures convergence. The PnP method IRCNN (Zhang et al., 2017b) uses a more lightweight CNN instead. While diffusion models are strong in image generation, DiffPIR (Zhu et al., 2023) cannot match DPIR in this superresolution task.

The two (bicubic for $s = 2$ and $s = 3$) end-to-end trained SwinIR transformer models (Liang et al., 2021) perform best in their noiseless training regime, but degrade significantly when the setting changes (see also Table 7 in Appendix C.1). There are no interpretable hyperparameters for adaptation, and retraining requires significantly more data compared to the minimal tuning of universal approaches. This lack of generalization is a major concern in practice.

### 5.3. MRI Reconstruction

Now, we deploy DEAL for MRI tasks. Specifically, we tackle the single- and 15-coil MRI setups detailed by Goujon et al. (2023). There, the ground truth consists of knee images from the fastMRI dataset (Knoll et al., 2020), both with fat suppression (PDFS) and without fat suppression (PD). The forward $\mathbf{H}$ involves $k$-fold subsampling in the Fourier domain and corruption by AGWN with $\sigma_\mathrm{n} = 0.002$. For each of the four evaluation tasks, we use ten images to tune the hyperparameters of all methods. In Table 3, we report the PSNR values on centered $(320 \times 320)$ patches of the remaining fifty test images. We compare against the popular TV regularization, the convex CRR regularizer (Goujon et al., 2023), its weakly convex extension WCRR, and the ProxDRUNet. All methods are universal and can be deployed without task-specific training. In Table 4, we report the computation times for several methods on a Tesla V100-SXM2-32GB GPU. We are significantly faster than the iterative refinement approach SAFI and get close to the (non-adaptive) WCRR baseline. Qualitative results are given in Figure 9 of Appendix C.3.

Table 2: Superresolution results on $(256 \times 256)$ center-cropped CBSD68 dataset. The number $\#\theta$ of parameters is given in millions. SwinIR is trained with a bicubic kernel and then also deployed for the others.

| | Category | $\#\theta$ | bicubic | 4 different kernels | | | bicubic | 4 different kernels | | |
| | | | PSNR (s = 2) | | | | PSNR (s=3) | | | |
| Noise $\sigma_\mathrm{n}$ | | | 0 | 2.55 | 7.65 | 12.75 | 0 | 2.55 | 7.65 | 12.75 |
|---|---|---|---|---|---|---|---|---|---|---|
| DEAL | Explicit Reg | 0.85 | 29.91 | **27.99** | 26.58 | 25.75 | 26.83 | **26.20** | 25.27 | 24.59 |
| Prox-PnP | Conv. PnP | 32.64 | - | 27.93 | **26.61** | 25.79 | - | 26.13 | **25.29** | **24.67** |
| IRCNN | PnP | **0.19** | 29.84 | 26.97 | 25.86 | 25.45 | 26.74 | 25.60 | 25.72 | 24.38 |
| DPIR | PnP | 32.64 | 29.63 | 27.79 | 26.58 | **25.83** | 26.70 | 26.05 | 25.27 | 24.66 |
| DiffPIR | Diff. Model | 93.56 | 29.73 | 27.84 | 26.48 | 25.63 | - | - | - | - |
| SwinIR | End to End | 11.75 | **30.88** | 24.56 | 22.84 | 20.73 | **27.76** | 22.41 | 21.24 | 19.53 |

## 5.4. Computational Scalability of DEAL

Here, we investigate the scalability of DEAL. To do so, we focus on a single-coil MRI experiment with 8-fold Cartesian masks of the size of the image, where only around 12 percent of the entries are nonzero. We use four images of sizes $256^2, 512^2, 1024^2$, and $2048^2$. The largest image is extracted from a high-resolution MRI brain image (Martinez et al., 2023). We perform our experiments on a Tesla V100-SXM2-32 GB GPU. In Table 5, we report the time and memory usage for each of the images. DEAL is consistently faster with a smaller memory footprint than Prox-DRUNet, which is the most similar method in terms of universality, performance, and convergence properties. Noteworthy, Prox-DRUNet fails for the image of size $2048^2$.

## 6. Interpretability and Robustness

We conclude empirically from Figures 4 and 10 that DEAL is not tied to a specific number $K_\mathrm{out}$ of iterations (6). In particular, doing more updates does not degrade the performance, unlike many PnP methods such as DPIR. The convergence to a fixed point (see Theorem 4.3) occurs for all experiments. In particular, both the relative error and the PSNR converge. To demonstrate the robustness regarding initialization, we instantiate the superresolution task. As we see in Figure 4, DEAL converges (in about 10 steps) to the same reconstruction, independently of the initialization, which is in accordance with Theorem 4.4. The relative errors for this task are given in Figure 11 of Appendix C.4, where the empirical convergence is emphasized once more.

We present visualizations for all parts of our architecture in Appendix D. Remarkably, we find mostly finite differences and their higher-order counterparts at various scales within $\mathbf{W}$ (see Figure 13 in the appendix). These filters extract the salient features of the input. The impulse response of $\mathbf{W}^\top \mathbf{W}$ and its Fourier transform are given in Figure 5. Empirically, we observe that $\ker(\mathbf{W}) = \mathrm{span}(\mathbf{1}_d)$ with $d$

Table 3: PSNR values for the MRI experiment.

| | 4-fold single coil | | 8-fold multi-coil | |
| | PD | PDFS | PD | PDFS |
|---|---|---|---|---|
| Zero-fill ($\mathbf{H}^\top y$) | 27.40 | 29.68 | 23.80 | 27.19 |
| TV | 32.44 | 32.67 | 32.77 | 33.38 |
| WCRR | 35.78 | 34.63 | 35.57 | 35.16 |
| SARR | 36.25 | 34.77 | 35.98 | 35.26 |
| SAFI | 36.43 | 34.92 | 36.06 | **35.36** |
| DEAL (Ours) | **36.59** | 34.92 | **36.21** | 35.32 |
| ProxDRUNet | 36.20 | **35.05** | 35.82 | 35.12 |
| PnP-DnCNN | 35.24 | 34.63 | 35.11 | 35.14 |

Table 4: Time (seconds) for the MRI experiment.

| | 4-fold single coil | | 8-fold multi-coil | |
| | PD | PDFS | PD | PDFS |
|---|---|---|---|---|
| WCRR | **12** | 20 | **9** | **8** |
| SAFI | 436 | 470 | 388 | 326 |
| DEAL (Ours) | 14 | **17** | 22 | 18 |
| ProxDRUNet | 113 | 38 | 170 | 105 |

as in (2). This is a practical certification for Proposition 4.1.

Our attention mechanism is multiplicative, see (4), rather than based on a traditional key-query parameterization. We present two striking interpretations of the mechanism. For simplicity, we focus on the castle-denoising example from Figure 2 with the final solution $\mathbf{x}_k$. In Figure 6, we illustrate two learned filters within $\mathbf{W}_c$ and the corresponding responses $\mathbf{W}_c\mathbf{x}_k$. The associated weights (masks) $\mathbf{m}_c(\mathbf{x}_k)$ are well-adapted to the structural features captured by these filters. In effect, the mask suppresses the image structures in the final squared responses $(\mathbf{m}_c(\mathbf{x}_k) \odot (\mathbf{w}_c * \mathbf{x}_k))^2$, which leads to a reduced regularization cost in (4). Hence, DEAL preserves salient structures. This is desirable as the image structure should not contribute to the cost.

| Method | $(256 \times 256)$ | | $(512 \times 512)$ | | $(1024 \times 1024)$ | | $(2048 \times 2048)$ | |
|---|---|---|---|---|---|---|---|---|
| | Time (s) | Mem. (GB) | Time (s) | Mem. (GB) | Time (s) | Mem. (GB) | Time (s) | Mem. (GB) |
| DEAL (Ours) | 6.4 | 0.38 | 36.0 | 1.51 | 173.0 | 6.03 | 1800.0 | 24.11 |
| Prox-DRUNet | 18.5 | 0.87 | 62.1 | 3.38 | 240.0 | 13.35 | NA | NA |

Table 5: Runtime and memory footprint in terms of different image sizes for MRI reconstruction.

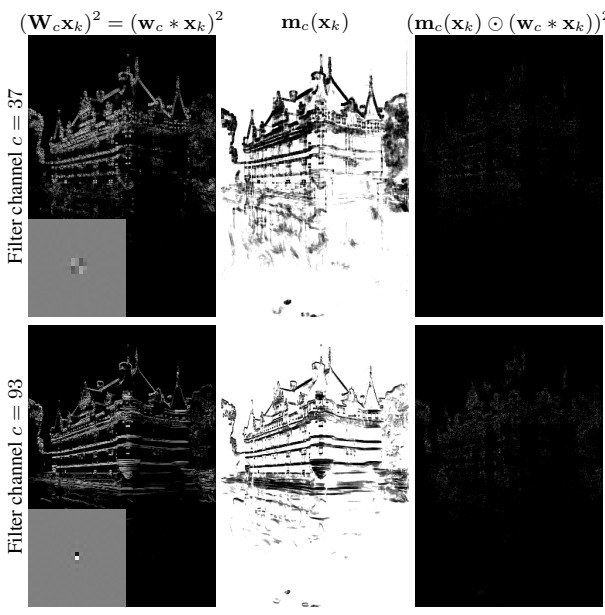

Figure 6: Learned filters. Left to right: Two filters $\mathbf{W}_c$ along with $(\mathbf{W}_c\mathbf{x}_k)^2$; corresponding masks $\mathbf{m}_c(\mathbf{x}_k)$; and adapted squared response.

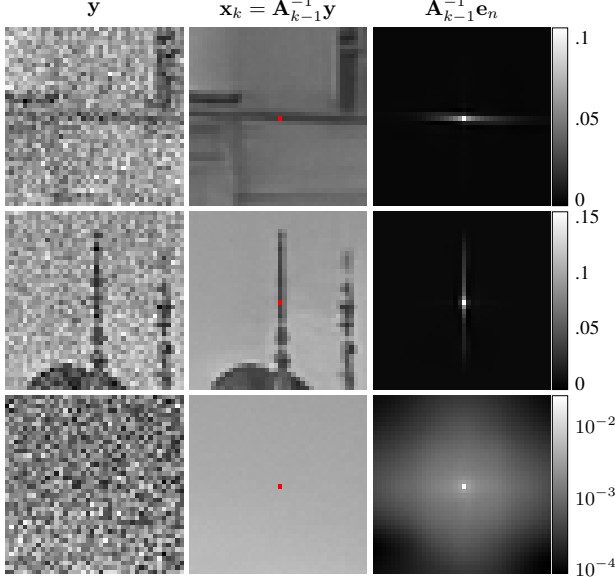

Figure 7: Adaptive-averaging interpretation. Left: Measurements $\mathbf{y}$. Middle: Reconstruction $\mathbf{x}_k$ with focus on its $n$th entry (in red). Right: Weight associated to $\mathbf{A}_{k-1}^{-1}$ at position $n$ with $\mathbf{x}_k[n] = \langle \mathbf{A}_{k-1}^{-1}\mathbf{e}_n, \mathbf{y}\rangle$. Top to bottom: Three regions of the castle image.

Next, we describe our second interpretation. Since we are in the denoising setting ($\mathbf{H} = \mathbf{I}$), we have that $\mathbf{x}_k = \mathbf{A}_{k-1}^{-1}\mathbf{y}$, with $\mathbf{A}_{k-1} = \mathbf{I} + \lambda \mathbf{W}^\top \mathbf{M}(\mathbf{x}_{k-1})^2\mathbf{W}$. Therefore, the mapping from the measurements $\mathbf{y}$ to the solution $\mathbf{x}_k$ is linear. More precisely, the $n$th component of the vectorized reconstruction $\mathbf{x}_k$ is a weighted average of the measurements $\mathbf{y}$ with the weight given by the $n$th row of $\mathbf{A}_{k-1}^{-1}$. To extract this row, we apply $\mathbf{A}_{k-1}^{-1}$ to the $n$th unit vector $\mathbf{e}_n$. In Figure 7, we see that $\mathbf{A}_{k-1}^{-1}\mathbf{e}_n$ aligns well with the structure of the neighborhood around the $n$th pixel. This indicates that spatial information is encoded into $\mathbf{A}_{k-1}$ during the refinements. For instance, directional averaging is observed in the first and second rows of Figure 7, while the image of its third row exhibits little structure and DEAL averages over a larger region, with an emphasis on the center pixel. Thus, at equilibrium, DEAL acts as an adaptive averaging mechanism, intelligently averaging the noisy measurements $\mathbf{y}$, with weights that emerge from our iterative refinements.

## 7. Conclusion

We have presented deep attentive least squares (DEAL) for image reconstruction. DEAL builds upon classic signal-processing ideas, which we blended with recent advances in deep learning, particularly, infinite-depth networks. It consists of two parts: (i) an iterative refinement of intermediate reconstructions based on a least-square-type problem; and (ii) a recurrent attention mechanism that adapts the problem spatially. We achieved competitive performance on different tasks while being able to provide interpretability, universality, and theoretical guarantees.

So far, we have only trained DEAL on a denoising task. If sufficient data are available, it appears possible to fine-tune all components of DEAL to further improve its performance. Moreover, DEAL is designed for the $\ell_2$ data fidelity, and devising extensions for other data-fidelity terms is an interesting direction of future work.

## Impact Statement

We aim to advance the field of image reconstruction by combining deep-learning tools with classic signal-processing techniques. We provide an architecture for the solution of

inverse problems that is interpretable and leads to competitive reconstruction quality. To the best of our knowledge, our work does not raise any ethical or societal concerns.

## Acknowledgements

M.P. and M.U. acknowledge support from the European Research Council (ERC Project FunLearn) under Grant 101020573 and by the Swiss National Science Foundation, Grant 200020_219356. S.N. and E.K. acknowledge support from the DFG within the SPP2298 under project number 543939932. E.K acknowledges support from the Austrian Science Fund (FWF) project number 10.55776/COE12.

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

## A. Ablation Studies

For the linear splines $\varphi_1$, $\varphi_2$, and $\varphi_3$ that appear in $\mathbf{M}$, we fix $r = 3$, $N_n = 31$. Moreover, we initialize $\varphi_1$ and $\varphi_2$ as the absolute value $|\cdot|$ and $\varphi_3$ as $\mathrm{e}^{-(\cdot)^2}$, motivated by classic anisotropic diffusion (Weickert, 1998). Each $s_c$ in (11) has 14 knots in the range $[-1, 51]$. They are initialized as the constant function 3. We choose $\sigma = \sigma_{\mathrm{n}}$ in $\mathbf{M}$, where $\sigma_{\mathrm{n}}$ is the standard deviation of the noise associated with a sample. We set $\lambda = \kappa(\sigma_{\mathrm{n}})$ with a learnable spline $\kappa$ defined using 52 knots in $[-1, 51]$ initialized as the identity. The denoisers are strongly tied, particularly, $\mathbf{W}$ and $\mathbf{M}$ must cope with various settings.

We used $N_{\mathrm{C}} = 128$ number of filters. We observed that a reduction of $N_{\mathrm{C}}$ to 64 or 32 degrades the performance for denoising on $\sigma = 25$ by around 0.1 and 0.15, respectively. We also observed that going beyond the filter size $(9 \times 9)$ in the Multi-Conv block does not improve the performance. The proposed constraints and initializations for the learning of the nonlinearities stem from the learning of such parts without any constraints and with zero initialization.

Since we used a shallow CNN for mask generation, we could not achieve good performance with conventional nonlinearities. Therefore, we use the deep-spline framework of (Bohra et al., 2020). We initially learned the splines with zero initialization and no constraints. There, we observed that $\varphi_1$ and $\varphi_2$ tended toward symmetric potentials, while the last $\phi_c$ resembled cutoff functions. These observations lead to our proposed configuration. The total number of parameters for all the learnable splines is fewer than 2000. Adding more spline knots (parameters) did not improve performance.

The attention mechanism is crucial for DEAL. Without it ($\mathbf{M} = \mathrm{Identity}$), the filter bank collapses to a single high-order Laplacian and leads to a huge drop in performance, as much as about 4dB in denoising for $\sigma_{\mathrm{n}} = 25$ with the grayscale images.

## B. Proofs

### B.1. Proof of Proposition 4.1

*Proof.* Assume that there exists $\mathbf{x} \in \mathbb{R}^d \setminus \{\mathbf{0}\} \in \ker(\mathbf{A}_k)$, so that $\mathbf{x}^\top \mathbf{A}_k \mathbf{x} = 0$. By definition of $\mathbf{A}_k$, this implies that $\mathbf{x}^\top \mathbf{H}^\top \mathbf{H} \mathbf{x} = \mathbf{0}$ and $\mathbf{x}^\top \mathbf{W}^\top \mathbf{M}^2(\mathbf{x}_k) \mathbf{W} \mathbf{x} = \mathbf{0}$. Hence, we get that $\mathbf{x} \in \ker(\mathbf{H})$ and $\mathbf{x} \in \ker(\mathbf{M}(\mathbf{x}_k)\mathbf{W})$, which is a contradiction. For $\mathbf{M}^2(\mathbf{x}_k) \succeq \epsilon_M \mathrm{Id}$, we estimate $\mathbf{A}_k \succeq \mathbf{H}^\top \mathbf{H} + \epsilon_M \mathbf{W}^\top \mathbf{W}$ and the uniqueness as in the first part. $\square$

### B.2. Proof of Lemma 4.2

*Proof.* Let $\hat{\mathbf{x}} = \mathcal{T}(\mathbf{x}, \mathbf{y}_1)$ and $\hat{\mathbf{z}} = \mathcal{T}(\mathbf{x}, \mathbf{y}_2)$, so that $\mathbf{A}_\mathbf{x}\hat{\mathbf{x}} = \mathbf{H}^\top \mathbf{y}_1$ and $\mathbf{A}_\mathbf{x}\hat{\mathbf{z}} = \mathbf{H}^\top \mathbf{y}_2$. This implies that $\mathbf{A}_\mathbf{x}(\hat{\mathbf{x}} - \hat{\mathbf{z}}) = \mathbf{H}^\top(\mathbf{y}_1 - \mathbf{y}_2)$ and we estimate

$$\|\hat{\mathbf{x}} - \hat{\mathbf{z}}\|_2 \leq \frac{\|\mathbf{H}\|_2}{\lambda_\epsilon} \|\mathbf{y}_1 - \mathbf{y}_2\|_2. \tag{19}$$

$\square$

### B.3. Proof of Theorem 4.3

*Proof.* First, we investigate the range of $\mathcal{T}(\cdot, \mathbf{y})$. By definition of $\mathcal{T}(\cdot, \mathbf{y})$, it holds for any $\mathbf{x} \in \mathbb{R}^d$ that

$$\|\mathcal{T}(\mathbf{x}, \mathbf{y})\|_2 = \|\mathbf{A}_k^{-1}\mathbf{H}^\top \mathbf{y}\|_2 \leq \frac{\|\mathbf{H}\mathbf{y}\|_2}{\lambda_\epsilon}. \tag{20}$$

For the second part, we want to apply the Brouwer fixed-point theorem. To this end, we must prove that $\mathcal{T}(\cdot, \mathbf{y})$ is continuous. Let $\mathbf{x}_1, \mathbf{x}_2 \in \mathbb{R}^d$, $\hat{\mathbf{x}}_1 = \mathcal{T}(\mathbf{x}_1, \mathbf{y})$, and $\hat{\mathbf{x}}_2 = \mathcal{T}(\mathbf{x}_2, \mathbf{y})$. Then, it holds that

$$\mathbf{A}_1 \hat{\mathbf{x}}_1 - \mathbf{A}_2 \hat{\mathbf{x}}_2 = \mathbf{0}$$
$$\mathbf{A}_1 \hat{\mathbf{x}}_1 - \mathbf{A}_1 \hat{\mathbf{x}}_2 = \mathbf{A}_2 \hat{\mathbf{x}}_2 - \mathbf{A}_1 \hat{\mathbf{x}}_2$$
$$\hat{\mathbf{x}}_1 - \hat{\mathbf{x}}_2 = \mathbf{A}_1^{-1}(\mathbf{A}_2 - \mathbf{A}_1)\hat{\mathbf{x}}_2. \tag{21}$$

Incorporating (20) and the normalization $\|\mathbf{W}\|_2 = 1$, we further infer that

$$\|\hat{\mathbf{x}}_1 - \hat{\mathbf{x}}_2\|_2 \leq \|\mathbf{A}_1^{-1}\|_2 \|\mathbf{W}\|_2^2 \|\mathbf{M}^2(\mathbf{x}_2) - \mathbf{M}^2(\mathbf{x}_1)\|_2 \|\hat{\mathbf{x}}_2\|_2$$
$$\leq L \frac{\|\mathbf{H}\mathbf{y}\|_2}{\lambda_\epsilon^2} \|\mathbf{x}_2 - \mathbf{x}_1\|_2. \tag{22}$$

Hence, $\mathcal{T}(\cdot, \mathbf{y})$ is Lipschitz-continuous and a fixed point exists. $\square$

### B.4. Proof of Theorem 4.4

*Proof.* Due to the Banach fixed-point theorem, the exponential convergence rate (17) holds. To estimate the difference of $\hat{\mathbf{x}} = \mathcal{T}(\hat{\mathbf{x}}, \mathbf{y}_1)$ and $\hat{\mathbf{z}} = \mathcal{T}(\hat{\mathbf{z}}, \mathbf{y}_2)$, we use the contractivity of $\mathcal{T}(\cdot, \mathbf{y}_1)$ and Lemma 4.2 to get that

$$\|\hat{\mathbf{x}} - \hat{\mathbf{y}}\|_2 = \|\mathcal{T}(\hat{\mathbf{x}}, \mathbf{y}_1) - \mathcal{T}(\hat{\mathbf{z}}, \mathbf{y}_2)\|_2 \leq \|\mathcal{T}(\hat{\mathbf{x}}, \mathbf{y}_1) - \mathcal{T}(\hat{\mathbf{z}}, \mathbf{y}_1)\|_2 + \|\mathcal{T}(\hat{\mathbf{z}}, \mathbf{y}_1) - \mathcal{T}(\hat{\mathbf{z}}, \mathbf{y}_2)\|_2$$

$$\leq q\|\hat{\mathbf{x}} - \hat{\mathbf{z}}\|_2 + \frac{\|\mathbf{H}\|_2}{\lambda_\epsilon}\|\mathbf{y}_1 - \mathbf{y}_2\|_2. \tag{23}$$

From this, we readily infer (18). □

## C. Additional Experiments

### C.1. Sensitivity to Hyperparameter Tuning and Generalization

Based on our observations in various setups, we always choose the model noise level $\sigma = 15$. In contrast, $\lambda$ must be adapted to the data noise level. This is typical for variational regularization, where $\lambda$ scales with the square of the data noise level $\sigma_n^2$. We investigate this closer for multi-coil MRI reconstruction at data noise level $\sigma_n = 0.002$. We give in Table 6 the reconstruction PSNR for different model noise levels $\sigma_n$ and regularization strengths $\lambda$. Indeed, the performance depends primarily on $\lambda$. Interestingly, even a tenfold change in $\lambda$ maintains a reasonable reconstruction quality. In general, a higher $\lambda$ leads to blurred images, while a lower $\lambda$ does not remove the artifacts.

Table 6: Hyperparameter Effect.

| $\sigma$ \ $\lambda$ | 0.01 | 0.1 | 1 | 10 |
|---|---|---|---|---|
| 5 | 27.0 | 33.77 | 32.25 | 29.28 |
| 15 | 32.5 | 33.88 | 31.8 | 28.31 |
| 25 | 33.05 | 33.61 | 31.42 | 27.75 |
| 50 | 32.96 | 33.05 | 30.96 | 26.75 |

Table 7: PNSR across kernels.

| Method | Bicubic | A | B | C | D |
|---|---|---|---|---|---|
| DEAL | 29.91 | 29.59 | 29.76 | 28.57 | 27.21 |
| SwinIR | 30.88 | 25.72 | 25.85 | 24.50 | 23.66 |

Regarding the adaptation of $\lambda$ to the noise level $\sigma_n$, we can also use a theoretically motivated closed-form formula for $\lambda$. As we see in Table 8, the performance drop is marginal compared to the fine-tuned case.

Table 8: Average reconstruction PSNR over 4 kernels for superresolution with $s = 2$ on centered-cropped CBSD68 data.

| Method | $\sigma_n = 2.55$ | $\sigma_n = 7.65$ | $\sigma_n = 12.75$ |
|---|---|---|---|
| DEAL (fine-tuned) | 27.99 | 26.58 | 25.75 |
| DEAL ($\lambda = 0.1 + 0.035\sigma_n^2$) | 27.97 | 26.57 | 25.75 |

To provide further evidence for the good generalization of DEAL, we perform superresolution with a downsampling rate of $s = 2$ without noise, where the SwinIR is trained on the bicubic kernel and $\lambda$ is fine-tuned on the bicubic task. Then, we apply SwinIR and DEAL to new kernels with no further change for both models. In Table 7, the kernels A-D are Gaussian kernels with different standard deviations (0.7, 1.2, 1.6, and 2.0). Then, the reconstruction PSNR on the center-cropped CBSD68 images is given. We conclude that SwinIR needs retraining for new kernels, while DEAL still performs well in the presence of a kernel mismatch with the hyperparameters being tuned on the bicubic kernel and deployed on the Gaussian ones.

## C.2. Grayscale Deblurring

Here, we evaluate the performance of the DEAL approach on a grayscale-deblurring task. We use the same setup as DPIR for this experiment (Zhang et al., 2022). This includes two blur kernels of sizes $(17 \times 17)$ and $(27 \times 27)$ from (Levin et al., 2009) and additive Gaussian noise with $\sigma_n = 2.55$ and $\sigma_n = 7.65$. In Table 9, we report the PSNR of the reconstructions for the Set3 images, namely Cameraman, House, and Monarch. We set the model noise level $\sigma = 15$ and $\lambda \in \{0.5, 2.5\}$ for the two given AWGN noise levels. We also compare with model-based EPLL (Zoran & Weiss, 2011) and the learning approach FDN that is specific to deblurring (Kruse et al., 2017). We observe that we are consistently the second-best method on this task after DPIR. We provide a visual comparison with DPIR in Figure 8.

Table 9: PSNR for grayscale deblurring.

| | $\sigma = 2.55$ | | | | | | $\sigma = 7.65$ | | | | | |
| | 17x17 | | | 27x27 | | | 17x17 | | | 27x27 | | |
| | C.man | House | Monarch | C.man | House | Monarch | C.man | House | Monarch | C.man | House | Monarch |
|---|---|---|---|---|---|---|---|---|---|---|---|---|
| EPLL | 29.18 | 32.33 | 27.32 | 27.85 | 28.13 | 22.92 | 24.82 | 28.50 | 23.03 | 24.31 | 26.02 | 20.86 |
| DEAL (Ours) | 31.72 | 35.20 | 32.77 | 31.64 | 35.03 | 32.48 | 27.89 | 32.24 | 28.26 | 27.79 | 32.11 | 28.15 |
| FDN | 29.09 | 29.75 | 29.13 | 28.78 | 29.29 | 28.60 | 26.18 | 28.01 | 25.86 | 26.13 | 27.41 | 25.39 |
| IRCNN | 31.69 | 35.04 | 32.71 | 31.56 | 34.73 | 32.42 | 27.70 | 31.94 | 28.23 | 27.58 | 31.55 | 27.99 |
| DPIR | **32.05** | **35.82** | **33.38** | **31.97** | **35.52** | **32.99** | **28.17** | **32.79** | **28.48** | **27.99** | **32.87** | **28.27** |

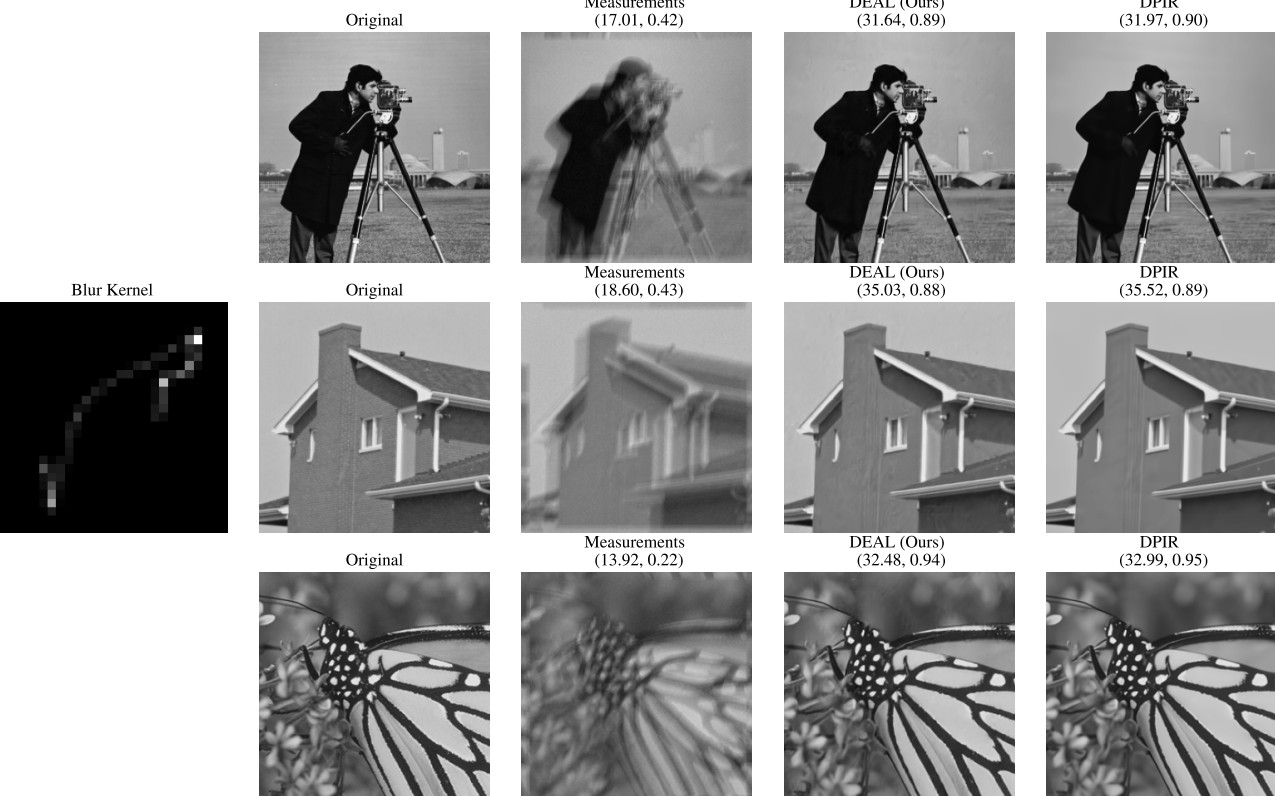

Figure 8: Both kernel and deblurring results for noise level $\sigma_n = 2.55$.

## C.3. Visualizations for MRI Reconstruction

We provide in Figure 9 visual reconstruction examples obtained with the different methods from Table 3. We also provide the solution path and the convergence plots for DEAL in Figure 10.

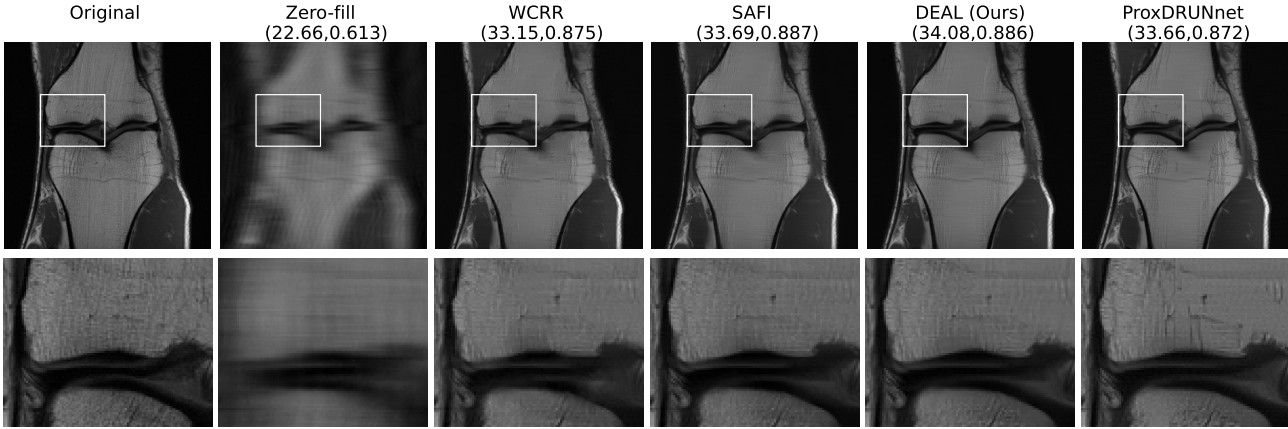

Figure 9: Eight-fold multi-coil MRI reconstructions for a PD image of the knee.

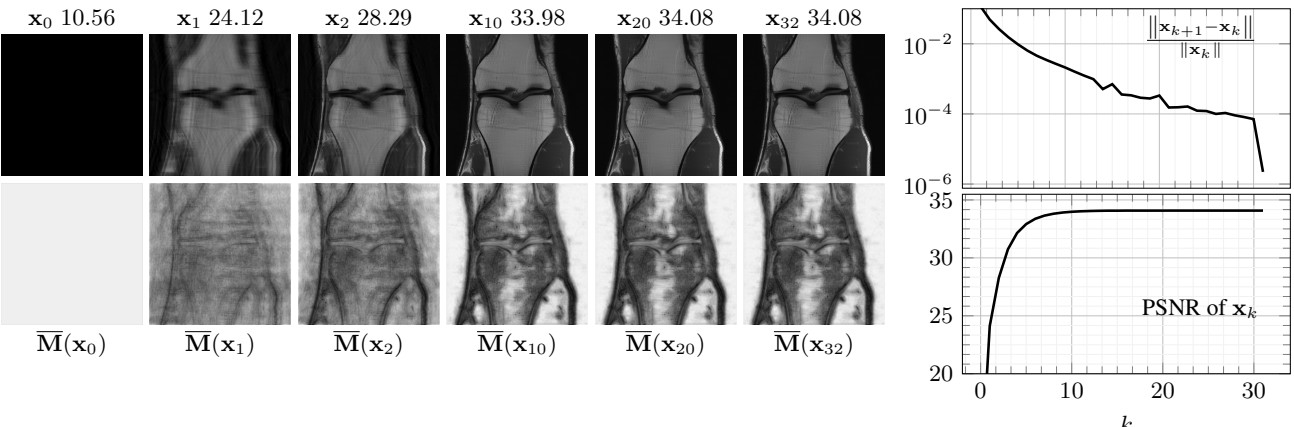

Figure 10: Solution and mask path for the 8-fold multi-coil MRI reconstruction for a PD image of the knee.

## C.4. Convergence Plot for Superresolution

In Figure 11, we represent the convergence plot for the superresoluion task on the setup of Figure 4.

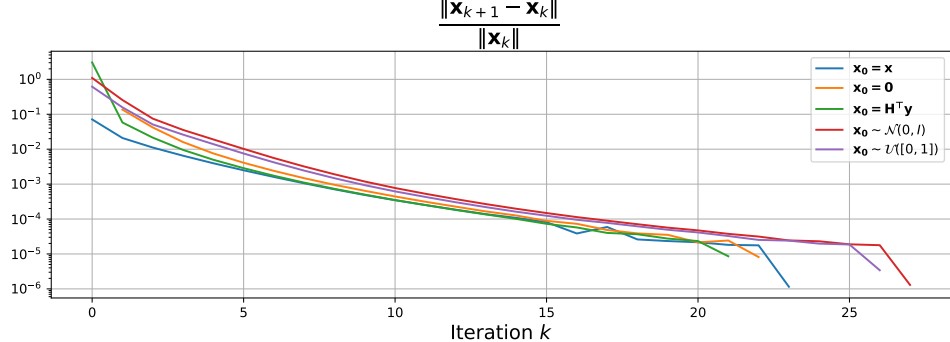

Figure 11: Convergence plot for the superresolution task with $s = 2$ and noise level $\sigma_\mathrm{n} = 2.55$ for different initializations.

## D. Visualization of Model Components

Now, we inspect the components of our learned grayscale model. In Figure 13, we depict the impulse response of $\mathbf{W}$, namely, the equivalent convolution kernels of the block(a.k.a filters). The convolutions of $\mathbf{W}_{\text{mask}}$ have similar structure, as we see in Figure 14. In addition, we depict the learned splines in Figure 12. For $\phi_c^\sigma$, we visualize three noise levels $\sigma \in \{5, 15, 25\}$ and channels $c \in \{44, 93, 99\}$. The three visualized channels correspond to vertical edge filters of various scales, see Figure 13. They resemble threshold functions that set high responses to zero. This results in less regularization in the regions that have high responses to the filters that are often activated by the image structures. This is a desirable behavior as the structure of the image should not contribute to the regularization cost. Additionally, the widths of the last spline $\phi_c^\sigma$ are increasing for all channels with respect to the noise level $\sigma$. Thus, more regularization is performed at higher noise levels. Moreover, we show the channel-wise average of the masks for the noisy and the denoised *castle* image in Figure 15. The masks remove the undesirable contribution of the image to the filter responses $\mathbf{Wx}$ of the regularizer. This results in lower penalization of the edges and yields sharper solutions.

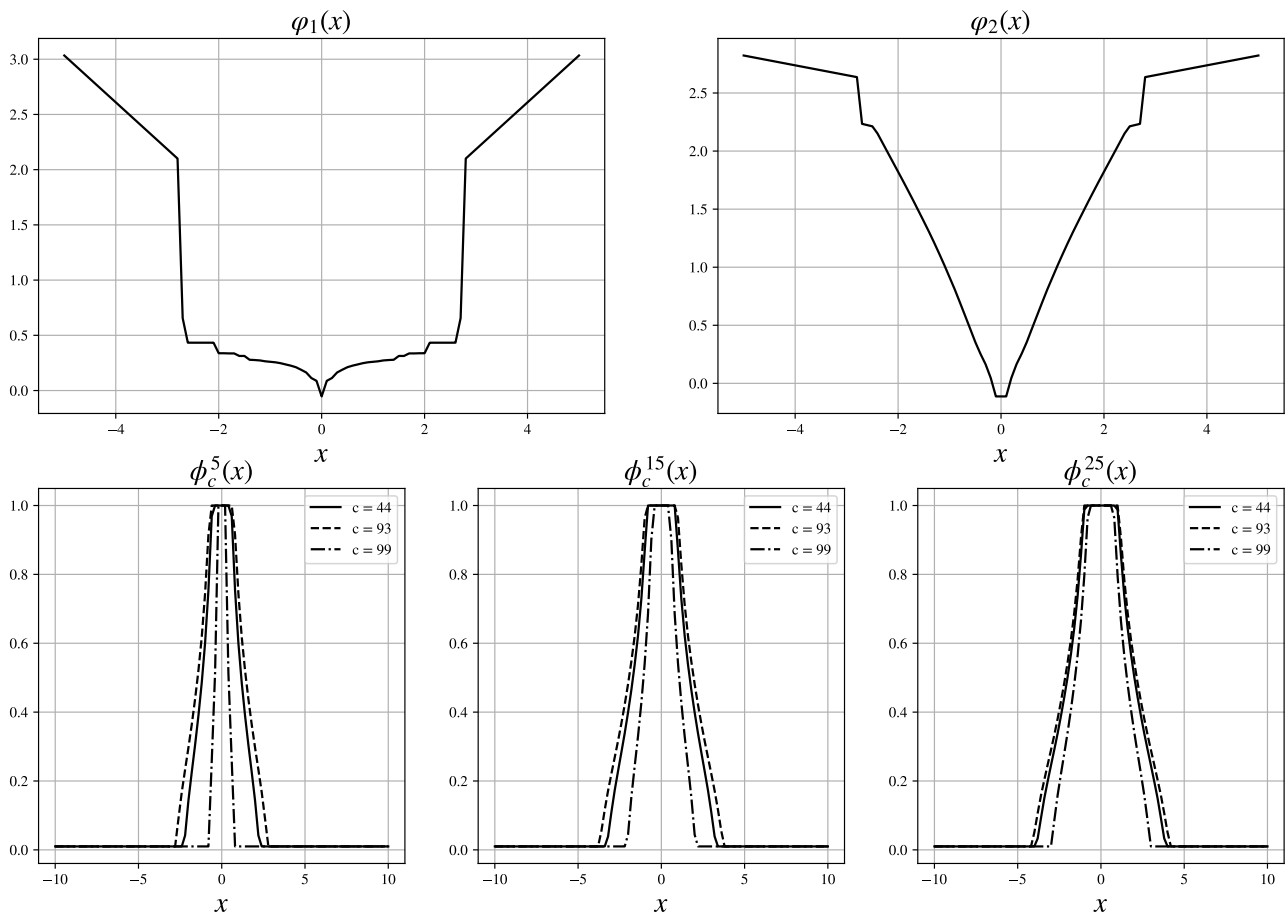

Figure 12: Learned splines in the mask-generation network $\mathbf{M}$ (Figure 1) for grayscale model.

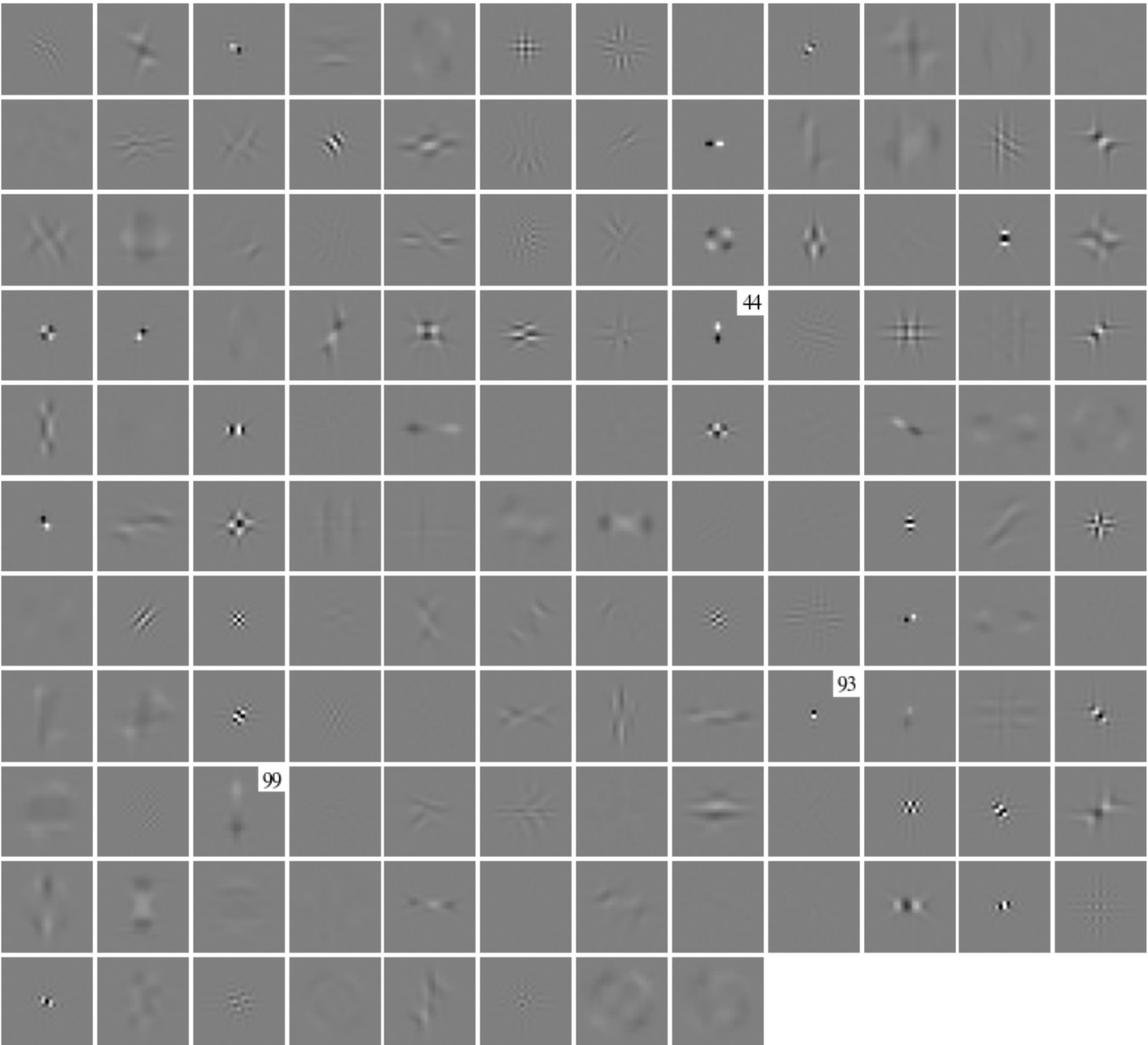

Figure 13: Effective convolution kernels for the Multi-Conv block $\mathbf{W}$ in the grayscale model. All plots use the same range, where neutral gray corresponds to zero.

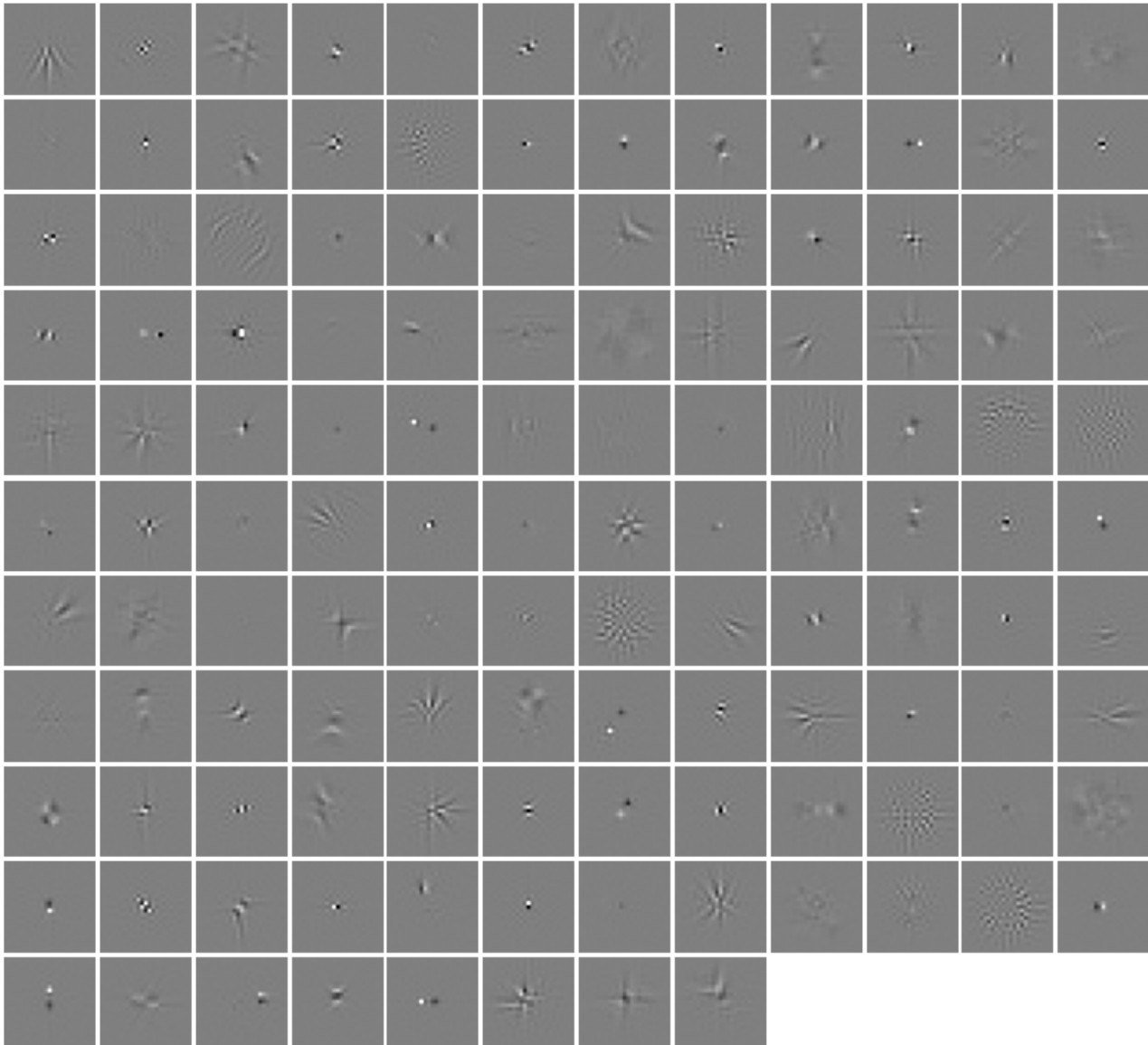

Figure 14: Effective convolution kernels for the Multi-Conv block $\mathbf{W}_{\mathrm{mask}}$ in the grayscale model. All plots use the same range, where neutral gray corresponds to zero.

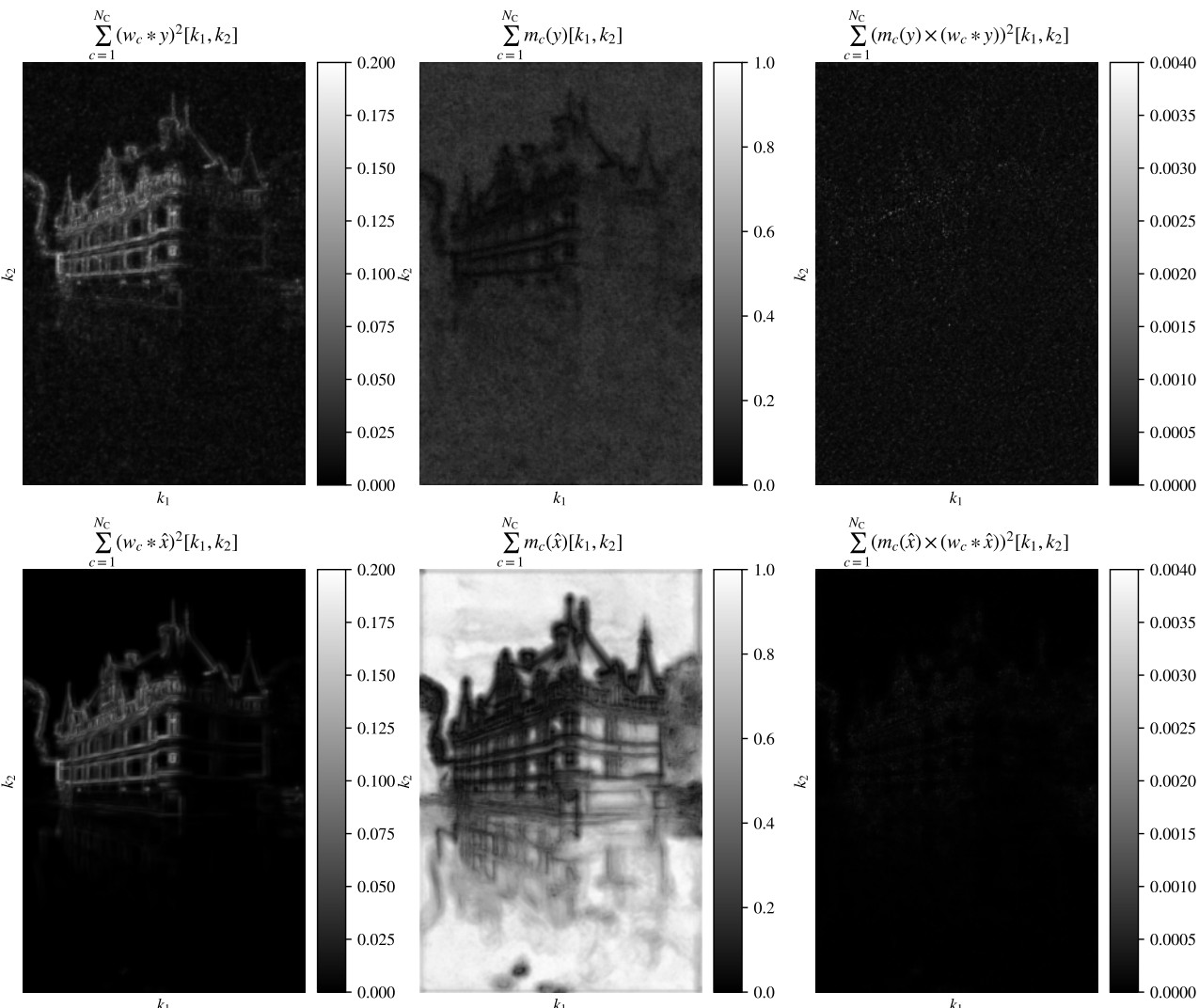

Figure 15: Filter responses and masks. From left to right: channel-wise average of (i) the squared response to the noisy image (top) and the solution of the castle denoising problem (bottom); (ii) the masks computed on the noisy image (top) and the solution (bottom); (iii) corresponding adapted responses.

