# OpenReview forum: "DEALing with Image Reconstruction: Deep Attentive Least Squares"
_ICML.cc/2025/Conference — ICML 2025 poster_

### Official Review · Reviewer_Q6VC · 2025-03-09

**Overall Recommendation:** 3

**Summary:**

This paper introduces Deep Attentive Least Squares (DEAL), a novel data-driven image reconstruction method that bridges traditional regularization techniques with modern deep learning. The authors propose an alternative to complex, highly parameterized deep architectures by leveraging the principles of classic Tikhonov regularization. The key idea is to iteratively refine intermediate reconstructions by solving a sequence of quadratic optimization problems. The method achieves state-of-the-art performance comparable to leading plug-and-play and learned regularizer approaches, while offering additional benefits such as interpretability, robustness, and provable convergence behavior.

## Update After Rebuttal

I would like to thank the authors for their detailed response and their efforts to address the issues raised in the initial review. I will maintain my recommendation to weak accept.

**Claims And Evidence:**

The claims made in the submission are well-supported by clear and convincing evidence, including theoretical proofs, extensive experiments, and comparisons with state-of-the-art methods.

**Essential References Not Discussed:**

While the paper provides a solid foundation, it could benefit from a more detailed discussion of Attention Mechanisms in Image Restoration in the related works section. For instance, citing seminal works such as "Attention-Guided CNN for Image Denoising" would help contextualize the proposed method within the broader landscape of attention-based approaches in image restoration.

**Experimental Designs Or Analyses:**

The experimental design and analyses are generally sound, but additional ablation studies and scalability tests could further strengthen the paper:
1. A more detailed analysis of the impact of the attention mechanism and spline parameterization
2. More results to show scalability of DEAL

**Methods And Evaluation Criteria:**

The proposed methods and evaluation criteria are appropriate and well-aligned with the problem of image reconstruction. The use of standard datasets, metrics, and comparisons with state-of-the-art methods ensures that the evaluation is rigorous and meaningful.

**Other Comments Or Suggestions:**

1. The font size in Figures 1 and 2 is somewhat small, which may hinder readability. Enlarging the font or providing higher-resolution versions of these figures would improve the reader's experience.
2. While the paper includes some ablation studies (e.g., filter size, number of filters), a more comprehensive analysis of the impact of different components (e.g., attention mechanism, spline parameterization) could further strengthen the claims.
3. Although DEAL demonstrates competitive performance, there is still a noticeable gap compared to methods like DRUNet (2022). To better highlight DEAL's advantages, the authors could include a table comparing the number of parameters, computational efficiency, and performance metrics (e.g., PSNR, SSIM) between DEAL and state-of-the-art methods. This would provide a clearer picture of DEAL's strengths, particularly in terms of efficiency and scalability.
4. Given the close connection between DEAL and attention mechanisms, it would be beneficial to add a dedicated section on attention mechanisms in image restoration in the related works. This would help situate DEAL within the broader context of attention-based approaches and highlight its unique contributions.

**Other Strengths And Weaknesses:**

Strengths: DEAL combines traditional regularization with deep learning in a novel way, this is a creative and original approach.  The paper is well-written and clearly presents the method, theoretical analysis, and experimental results. The visualizations and supplementary material enhance the clarity of the presentation.

Weakness: Some details could be further refined and supplemented.

**Questions For Authors:**

While DEAL is shown to be efficient for moderate-sized problems, I’m curious about its performance on very large-scale datasets (e.g., high-resolution medical images).

**Relation To Broader Scientific Literature:**

DEAL's contributions are well-grounded in the broader literature, building on and extending prior work in iterative refinement, learned regularization, and theoretical analysis. The novel idea of iterative refinement with learned filters and an attention mechanism is particularly noteworthy and has the potential to inspire future research in this area.

**Theoretical Claims:**

The theoretical claims are well-supported by correct proofs, the authors provide a rigorous theoretical foundation for their method. The theoretical claims like Proposition 4.1 (uniqueness of updates), Lemma 4.2 (Lipschitz continuity), Theorem 4.3 (existence of fixed points), and Theorem 4.4 (exponential convergence) are supported by standard mathematical tools such as positive definiteness, Lipschitz continuity, and the Banach fixed-point theorem. These claims are thoroughly explained in both the main text and the supplementary material.  Some assumptions, like ker(H)∩ker(W)=0 in the manuscript are reasonable and empirically supported too.

---

> ### Author Rebuttal · Authors · 2025-03-31
>
> We appreciate the reviewer’s time and feedback and are glad that DEAL is recognized as creative and original. To summarize our responses:
> - We added a paragraph to the related work focusing on attention mechanisms.
> - We performed a new experiment to survey the scalability of DEAL for large medical images.
> - We provided more ablation on the impact of the attention mechanism.
>
> > 1. The font size in Figures 1 and 2 is somewhat small...
>
> We will enlarge the font-sizes and improve the resolution of the Figures.
>
> > 2. While the paper includes some ablation studies ...
>
> To highlight the impact of the attention mechanism and the spline parameterization, we added the following ablation:
>
> - The attention mechanism is crucial for DEAL. Without it (M= Identity), the filter bank collapses to a single high-order Laplacian, leading to a huge performance drop e.g. around 4dB in denoising for $\sigma_n =25$ on our grayscale testset.
>
> - Since we use a shallow CNN for mask generation, we cannot achieve good performance with conventional nonlinearities. Therefore, we use the deep spline framework of [1]. We initially learned the splines with zero initialization and no constraints. There, we observed that $\varphi_1$ and $\varphi_2$ tended toward symmetric potentials, while the last $\{\phi_c\}$ resembled cut-off functions. These observations lead to our proposed setup for these splines as described in the paper. The total number of the parameters for all the learnable splines is less than 2000 and adding more spline knots (parameters) does not improve performance.
>
> > 3. Although DEAL demonstrates competitive performance ...
>
> We augmented our tables and added more metrics including the number of parameters to further clarify our comparisons with SOTA methods (see the updated table in point 2 reviewer dGc6). We also provide a new large-scale experiment for the MRI setup to showcase the scalability of DEAL (see response point [Questions For Authors]).
>
> > 4. Given the close connection between DEAL and attention mechanisms...
>
> We will add this paragraph to the related work section:
> **Attention Mechanism**: Originally popular in natural language processing, attention mechanisms are now widely used in image processing. Two main approaches exist: (i) patch-based attention, where images are divided into patches and processed using scaled dot-product attention, as in SwinIR [2] and Restormer [3]; and (ii) point-wise multiplicative attention, which integrates attention directly into the architecture, as in attention-guided CNNs [4]. While patch-based methods effectively capture long-range dependencies, they introduce significant computational overhead due to the need to compute inner products across all patches in a latent space. In contrast, point-wise attention avoids this, making it a more efficient alternative. Our approach to implementing attention is more aligned with methods that rely on point-wise multiplication, rather than the key-query parameterization of transformers. In particular, the way attention is incorporated in DEAL is highly interpretable (see Section 6).
>
> We will add more references to the paper's version and adapt the writing of Section 6 to emphasize more on the interpretability of our attention mechanism.
>
> > [Questions For Authors]: While DEAL is shown to be efficient for moderate-sized problems...
>
> We designed a new MRI experiment to address the questions. We use four images of sizes 265x256, 512x512, 1024x1024 and 2048x2048. The largest one is extracted from a high-resolution MRI brain image [5]. The smaller ones are obtained by bicubic downsampling. For the MRI setup, we use 8-fold Cartesian masks of the size of the image, where only around 12 percent of the mask entries are non-zero. We perform our experiments on a TeslaV100-SXM2-32GB GPU. There, DEAL can successfully handle the reconstruction of an image of size 2048x2048. In contrast, Prox-DRUNet fails at this size. To provide more insights, we add the following table in the paper, where we report the time and memory usage for each of the images in the aforementioned setup. DEAL is consistently faster than Prox-DRUNet with less memory usage.
>
> | Method| 256x256 (Time s)| 256x256 (Mem. GB)| 512x512 (Time s)|512x512 (Mem. GB)|1024x1024 (Time s)| 1024x1024 (Mem. GB)|2048x2048 (Time s)|2048x2048 (Mem. GB)|
> |-|-|-|-|-|-|-|-|-|
> |DEAL (Ours)|6.4|0.38|36.0|1.51|173.0|6.03|1800.0|24.11|
> |Prox-DRUNet|18.5|0.87|62.1|3.38|240.0|13.35|NA|NA|
>
> Refs.
>
> [1] Bohra et al. "Learning activation functions in deep (spline) neural networks.", 2020.
>
> [2] Liang et al. "Swinir: Image restoration using swin transformer.", 2021.
>
> [3] Zamir et al. "Restormer: Efficient transformer for high-resolution image restoration.", 2022.
>
> [4] Tian et al. "Attention-guided CNN for image denoising." 2020.
>
> [5] Martinez et al. "BigBrain-MR: a new digital phantom with anatomically-realistic magnetic resonance properties at 100-µm resolution for magnetic resonance methods development." 2023.

---

### Official Review · Reviewer_on6j · 2025-03-14

**Overall Recommendation:** 4

**Summary:**

This paper presents a least-square-type image reconstruction method. It is formulated as a conjugate gradient method. It consists of of an iterative refinement process with two main components: one that estimates the reconstructed image and a another that generates a mask, modulating the response of the prior filter in a spatially adaptive manner for its following iterations. Similar to plug-and-play methods, the proposed model is trained on the denoising task and can be universally applied to other image reconstruction tasks given the forward operator and a validation set on the target task to set its hyperparameters, including the standard deviation of noise and the weight assigned to the prior term. The strength of the proposed method is two fold. It achieves near SOTA performance comparable to similarly universally pretrained methods with a relatively low computational complexity, while also ensuring robustness and convergence to a unique solution.

## update after rebuttal
I thank the authors for responding to my main concerns. I will maintain my recommendation to accept.

**Claims And Evidence:**

- Proof for theoretical claims are provided in the supplementary material.

- Convergence: The authors refer to Theorem 4.3 and illustrate examples on how convergence to a fixed point occurs for both the relative error and the resulting PSNR of the reconstruction.

- Robustness: An example of robustness to various forms of initialization for the 2x super-resolution task is provided.

- Interpretability: The authors present a couple of examples of learned filters for the denoising task and offer reasonable interpretations of how the associated masks are adapted to the structural features captured by those filters.

- Performance: Near SOTA results are demonstrated on multiple tasks including, denoising, super-resolution and MRI.

**Essential References Not Discussed:**

N.A.

**Experimental Designs Or Analyses:**

Regarding the super-resolution task, would it be fare to expect the compared methods to account for the considered addition of noise?

**Methods And Evaluation Criteria:**

Yes, they are commonly considered evaluation criteria in the literature.

**Other Comments Or Suggestions:**

No other comments.

**Other Strengths And Weaknesses:**

- Originality: The particular approach to formulate the problem and solve it with confidence in convergence to a unique solution seems original.

- Clarity: The provided illustrations for the experiments clarify the claims, at least for a the limited cases considered.

- Significant: Based on the identified related works, the proposed methods is a significant improvement in terms of the trade-off on performance and efficiency, while maintaining theoretical convergence properties.

**Questions For Authors:**

1. The authors have chosen only a handful of baselines for the comparisons. Why those specific methods and why only those and not others?

2. How does the performance compare to SOTA deep learning based methods without theoretical guarantees, such as transformers and diffusion-based methods that claim to solve multiple tasks including those considered in the experiments?

3. Is it reasonable to also test the performance without searching for the considered hyperparameters? In other words, as an additional baseline, how bad does the performance get without access to a validation set?

**Relation To Broader Scientific Literature:**

The proposed least-square-type image reconstruction method builds on classical regularizers that focus on sparsity in image gradients and parametric models. The method utilizes implicit regularization techniques using plug-and-play denoisers and spatial adaptivity, which are allow it to achieved improved performance. Iterative refinement methods and nonlocal Laplacians contribute to its robustness and convergence. Presenting near state-of-the-art performance with low computational complexity makes it well-positioned among efficient neural network architectures for image reconstruction.

**Theoretical Claims:**

I did not go through all the proofs provided in the supplementary material.

---

> ### Author Rebuttal · Authors · 2025-03-31
>
> We appreciate the reviewer's time and feedback and are glad that DEAL is recognized as a significant improvement in the performance-efficiency trade-off within image reconstruction techniques. In summary,
> - We clarified our choice of comparison methods.
> - We added diffusion models and transformers as additional comparisons for super-resolution.
> - We provided insights into hyperparameter sensitivity.
>
> >  [Experimental Designs Or Analyses] Regarding the super-resolution task, would it be fare to expect the compared methods to account for the considered addition of noise?
>
> For super-resolution, we use the exact same setup as the ProxDRUNet and use their given optimal hyperparameters that are different for each noise level. For IRCNN and DPIR, formulas for optimal hyperparameters are given within their codes based on the noise level for the super-resolution task. For MRI, we follow the WCRR paper's hyperparameter tuning setup with ten validation images to adapt all methods.
>
> > 1. Why those specific methods and why only those and not others?
>
> The competing methods are selected based on code availability and similar experimental setups. Due to their flexibility, and since our method is also universal, we mainly focused on PnP approaches.
>
> Super-resolution: Here, DPIR is the SOTA among PnP methods. DPIR is tied with a fix number of steps and its performance will degrade if iterated more. By constraining the underlying DRUnet, ProxDRUnet addresses this problem and obtains convergence guarantees. Instead of a DRUnet, IRCNN deploys a lightweight CNN. Now, we added the end-to-end trained transformer SwinIR [1] and the diffusion model DiffPIR [2].
>
> MRI: For medical grayscale images, we also compare with classical TV and the SOTA explicit learned regularization methods WCRR, SARR, and SAFI.
>
> > 2. How does the performance compare to SOTA deep learning-based methods without theoretical guarantees...
>
>  We added comparisons to the transformer-based SwinIR and diffusion models for superresolution. End-to-end models like SwinIR are highly sensitive to their training tasks and degrade significantly when conditions change. Diffusion models excel in image generation, but the current diffusion-based reconstruction methods often perform worse than DPIR for image super-resolution in terms of PSNR metric (see Table 2 of the DiffPIR paper [3]). Our superresolution results confirm these findings (see response point 2 of reviewer dGc6 for the updated table).
>
>
> > 3. Is it reasonable to also test the performance without searching for the considered hyperparameters? In other words, as an additional baseline, how bad does the performance get without access to a validation set?
>
>  Regarding the two hyperparameters, we can always set the model noise level $ \sigma = 15 $ based on our observations in different setups. In contrast, $\lambda$ must be adapted to the data noise level. This is typical for variational regularization, where $\lambda$ inversely scales with the data noise level $\sigma_n^2$. We investigate this closer for multi-coil MRI reconstruction at data noise level $\sigma_n = 0.002$. Indeed, the performance depends primarily on $\lambda$. Interestingly, even a 10-fold change in $\lambda$ maintains reasonable reconstruction quality. In general, higher $\lambda$ leads to blurred images, while lower $\lambda$ does not remove the artifacts.
>
>  In the following table, for the mentioned MRI setup, the reconstruction PSNR for the different choices of the model noise level $\sigma$ and regularization strength $\lambda$ is given.
> | σ \ λ|0.01|0.1|1|10|
> |-|-|-|-|-|
> | 5|27.00|33.77|32.25|29.28|
> |15|32.50|33.88|31.80|28.31|
> |25|33.05|33.61|31.42|27.75|
> |50|32.96|33.05|30.96|26.75|
>
>
> Refs.
>
> [1] Liang et al. "Swinir: Image restoration using swin transformer.", 2021.
>
> [2] Zhu et al. "Denoising diffusion models for plug-and-play image restoration.", 2023.

---

### Official Review · Reviewer_yde4 · 2025-03-16

**Overall Recommendation:** 3

**Summary:**

This paper presents Deep Attentive Least Squares (DEAL), a novel image reconstruction method that bridges traditional signal processing and modern deep learning. DEAL formulates reconstruction as an iterative least squares problem with spatially adaptive regularization, where an attention mechanism dynamically modulates the regularization weight based on the local image structure. The approach efficiently solves a sequence of quadratic problems using a learned multi-convolution filter and a conjugate gradient solver, ensuring both interpretability and computational efficiency. The authors provide rigorous theoretical analysis, proving the uniqueness of solutions, convergence guarantees, and robustness to initialization. Experimentally, DEAL is evaluated on denoising, super-resolution, and MRI reconstruction, achieving competitive results while using significantly fewer parameters than state-of-the-art deep learning models. The study highlights DEAL’s interpretability, universality, and theoretical guarantees, positioning it as a promising alternative to heavily parameterized deep models for inverse imaging problems.

**Claims And Evidence:**

1. Claim: DEAL provides a principled approach to image reconstruction, combining classic signal processing techniques with deep learning insights.
Evidence: DEAL is formulated as an iterative least-squares problem with spatially adaptive regularization. The authors draw connections to traditional Tikhonov regularization and modern plug-and-play (PnP) methods, demonstrating how DEAL balances interpretability and performance.

2. Claim: DEAL achieves convergence to a unique fixed point, ensuring stability and robustness.
Evidence: The authors provide rigorous theoretical guarantees (Propositions 4.1–4.4), proving uniqueness under mild conditions and showing that the iterative updates lead to a contraction mapping under certain assumptions. Empirically, DEAL consistently converges in experiments, regardless of initialization.

3. Claim: DEAL outperforms traditional spatially adaptive regularization methods and achieves performance close to deep learning-based approaches while using significantly fewer parameters.
Evidence: Extensive experiments on denoising, super-resolution, and MRI reconstruction show that DEAL surpasses non-adaptive regularization methods like WCRR and SARR. Furthermore, Table 1 and Table 2 demonstrate that DEAL’s PSNR scores approach those of DRUNet-based models, despite having 30× fewer parameters.

4. Claim: The learned attention mechanism effectively suppresses regularization in structured image regions, preserving details.
Evidence: The authors provide visualizations (Figures 7 and 8), showing how learned masks adapt to image structures. The masks reduce regularization in smooth areas while enforcing it in noisy regions, preventing excessive smoothing of important features.

5. Claim: DEAL is robust to hyperparameters and initialization, making it a reliable reconstruction method.
Evidence: Experiments in Figure 5 and Appendix C.3 show that DEAL consistently converges to the same solution regardless of initialization. The model is also shown to be less sensitive to hyperparameter tuning compared to deep learning-based methods.

6. Claim: DEAL generalizes well across different inverse problems without task-specific retraining.
Evidence: Unlike deep neural networks that require retraining for each task, DEAL is directly applied to denoising, super-resolution, and MRI reconstruction by simply adjusting two hyperparameters (σandλ). The results in Table 3 and Table 4 confirm its versatility.

The claims in this paper are well-supported by both mathematical theory and empirical validation. The combination of theoretical convergence guarantees, extensive benchmarking, and qualitative visualizations strongly substantiates DEAL’s effectiveness and robustness for inverse imaging problems

**Essential References Not Discussed:**

The paper does not overlook any critical related research, and all core references relevant to the DEAL method are appropriately cited. The authors correctly reference key works in the field of image reconstruction and regularization, such as the fields-of-experts model and PnP methods, and do not neglect any prior research crucial to this work. So the paper properly attributes existing methods and cites relevant literature without significant omissions.

**Experimental Designs Or Analyses:**

The experimental design in this paper is well-structured and aligned with the research objectives, ensuring a comprehensive evaluation of DEAL across multiple inverse imaging tasks. Below is an assessment of the key experimental components:

1、The study evaluates DEAL on denoising, super-resolution, and MRI reconstruction, aligning well with its iterative least squares formulation and spatially adaptive regularization framework.

2、The authors use BSD68, CBSD68, and fastMRI datasets and compare DEAL against BM3D, TV, WCRR, SARR, SAFI, DnCNN, DRUNet, and Prox-DRUNet, ensuring a fair and comprehensive benchmark.

3、DEAL is trained on an AWGN denoising task with σ_n∈[0, 50], using Adam optimization and cosine annealing, but lacks detailed sensitivity analysis on hyperparameters.

4、The results are quantitatively evaluated using PSNR (Tables 1-3), SSIM (Figures 3 & 5), and convergence analysis (Figure 4), but statistical significance tests (e.g., confidence intervals) are not reported.

5、The study provides clear training details and evaluation protocols, demonstrating robustness across different conditions, but code availability is not explicitly mentioned.

**Methods And Evaluation Criteria:**

Methods：
The authors of this paper propose the Deep Attentive Least Squares (DEAL) method to address the image reconstruction problem, integrating classical regularization techniques with modern deep learning approaches:

1. Formulation as an Iterative Least Squares Optimization with Adaptive Regularization

DEAL formulates the image reconstruction problem as an iterative least squares optimization incorporating a spatially adaptive regularization term. In each iteration, a conjugate gradient (CG) algorithm is employed to solve the quadratic optimization problem efficiently. The regularization term is dynamically adjusted via learned attention weights, allowing the model to adapt to different image regions and enhance reconstruction quality.

2. Spatially Adaptive Regularization via an Attention Mechanism

A shallow convolutional neural network (CNN) is used to compute attention weights, which modulate the local impact of the regularization term. The CNN extracts multi-scale features and applies pointwise nonlinear transformations to learn an adaptive weight distribution. This learned attention mechanism allows DEAL to reduce regularization in structured regions while enhancing regularization in noisy areas, thereby preserving critical image details.

3. Multi-Convolution Module for Efficient Feature Extraction

DEAL employs a multi-layer convolutional structure to extract spatial features efficiently. These convolutional layers do not incorporate nonlinear activation functions, ensuring a large receptive field while maintaining interpretability. This architecture is used in both the reconstruction process and the mask generation module, allowing for shared feature representation.

4. Training on a Denoising Task to Improve Generalization

To enhance generalization, DEAL is trained on an additive white Gaussian noise (AWGN) denoising task, where the noise level is set as
σ_n∈[0,50]. The training data includes both grayscale and color images, utilizing the BSD68 and CBSD68 datasets. The loss function consists of three components:
①Mean Squared Error (MSE): Ensures consistency between the reconstructed output and the ground truth image.
②Stability Penalty on Attention Weights: Suppresses unstable variations in the regularization adjustment.
③Total Variation Regularization on Learned Activation Functions: Prevents excessive complexity in the nonlinear transformations.

5. Applicability to Various Inverse Problems

Once trained, DEAL can be directly applied to a range of inverse problems, including denoising, super-resolution, and MRI reconstruction, without requiring additional training. The model only requires tuning of two hyperparameters—regularization strengthλand noise levelσ to adapt to different tasks.

Evaluation Criteria

The authors evaluate DEAL using both quantitative and qualitative metrics, comparing its performance across multiple inverse imaging tasks.

1. PSNR

PSNR is the primary metric used to assess image quality and is widely applied in inverse imaging tasks.A higher PSNR value indicates a reconstruction closer to the ground truth image.Tables 1, 2, and 3 report the PSNR results of DEAL for denoising, super-resolution, and MRI reconstruction.

2. SSIM

SSIM measures the structural similarity between the reconstructed and ground truth images, evaluating perceptual quality.Figures 3 and 5 demonstrate that DEAL preserves fine details better than competing methods.

3. Convergence and Stability

Theoretical analysis (Theorems 4.3 and 4.4) proves that DEAL has a unique solution and guarantees convergence.Experimental results in Figure 4 show that DEAL consistently converges under different initialization conditions.Unlike Plug-and-Play (PnP) approaches (e.g., DPIR), where performance may degrade with excessive iterations, DEAL steadily converges to a fixed point, ensuring stable performance.

4. Comparison with State-of-the-Art Methods

DEAL is compared against the following methods:
①Classic methods: BM3D, Total Variation (TV) regularization.②Learned regularization approaches: WCRR, SARR, SAFI.③Deep learning models: DnCNN, DRUNet, Prox-DRUNet.
Tables 1, 2, and 3 show that DEAL achieves performance close to DRUNet-based methods while using only 1/30 of the parameters, highlighting its advantage in computational efficiency.

5. Computational Efficiency

Table 4 reports the computational time for MRI reconstruction, showing that DEAL is significantly faster than the iterative refinement method SAFI and comparable to the non-adaptive WCRR baseline.Figure 5 further illustrates DEAL’s efficiency in super-resolution tasks, demonstrating faster and more stable convergence compared to PnP-based methods.

The authors provide rigorous theoretical guarantees for Deep Attentive Least Squares (DEAL), demonstrating its uniqueness, convergence properties, and stability under certain conditions. The theoretical foundation is based on iterative least squares optimization with spatially adaptive regularization.

**Other Comments Or Suggestions:**

N/A

**Other Strengths And Weaknesses:**

Strengths:
The paper excels in originality by proposing DEAL, a novel approach that effectively combines traditional signal processing with modern deep learning techniques, providing a clear solution to challenges in image reconstruction. The method’s theoretical guarantees for convergence and robustness add significant value, making it stand out among other methods in the field. The contributions are highly significant, offering improvements in interpretability, computational efficiency, and performance. The paper’s clarity is strong, with well-defined methodology, clear explanations, and comprehensive experimental results that demonstrate the efficacy of the approach.

Weaknesses:
While the contributions are important, there is a slight limitation in the generalization of the method to other tasks beyond the presented inverse problems, such as super-resolution or more complex imaging scenarios. Additionally, although the paper is well-written, some aspects could benefit from further elaboration, such as the details of the learned attention mechanism and its practical applications in different domains. Despite this, the overall structure and writing are clear and accessible, making the paper easy to follow.

**Questions For Authors:**

N/A

**Relation To Broader Scientific Literature:**

The contributions of this paper are closely connected to existing research in the field of image reconstruction. The authors propose the DEAL method, which combines the fields-of-experts model (Roth & Black, 2009) and PnP methods (Venkatakrishnan et al., 2013), integrating traditional signal processing and modern deep learning techniques to introduce a novel image reconstruction method. Compared to previous learning-based regularization methods, the proposed DEAL method offers significant advantages in terms of interpretability, computational efficiency, and convergence.
The paper correctly cites relevant literature, covering important studies in image reconstruction and regularization, such as the learning paradigms discussed by Chen et al., 2014 and Effland et al., 2020, as well as the parameterization strategies addressed by Goujon et al., 2023 and Zach et al., 2024. The authors also reference research on autoencoders (Li et al., 2020) and algorithm unrolling (Kobler et al., 2022), accurately highlighting the relationship and distinctions between DEAL and these methods.
In terms of innovation and impact, the paper contributes by combining classical signal processing techniques with deep learning methods, proposing a new approach that integrates iterative refinement and attention mechanisms. Unlike deep learning methods like DRUNet, DEAL provides convergence and robustness guarantees, with clear advantages in computational efficiency and interpretability. The innovative combination in DEAL positions it with broad potential applications and influence in the field of image reconstruction.
Overall, the paper correctly cites relevant literature and proposes an innovative and promising solution in the field of image reconstruction.

**Theoretical Claims:**

The authors provide rigorous theoretical guarantees for Deep Attentive Least Squares (DEAL), demonstrating its uniqueness, convergence properties, and stability under certain conditions. The theoretical foundation is based on iterative least squares optimization with spatially adaptive regularization.

1、Uniqueness of the Solution
Claim: DEAL ensures a unique solution at each iteration, provided that the problem satisfies specific conditions.
Supporting Theory: Proposition 4.1 states that if the intersection of the null spaces of H and M(x_k)W contains only the zero vector, i.e., ker(H)∩ker(M(x_k)W) = {0}, then the system matrix A_k is positive definite, ensuring the uniqueness of the solution in equation (8).

2、Lipschitz Continuity and Stability of the Iterative Process
Claim: The iterative update operator T(x, y) is Lipschitz continuous, ensuring stability across iterations.
Supporting Theory: Lemma 4.2 proves that if ker(H) ∩ ker(W) = {0} and M(x)^2 is greater than or equal to epsilon_M times the identity matrix, then T(x, y) is Lipschitz continuous with a bounded Lipschitz constant.

3、Existence of a Fixed Point
Claim: DEAL converges to a fixed point, ensuring stability and reliability in practical applications.
Supporting Theory: Theorem 4.3 establishes that T(x, y) maps into a bounded region and has at least one fixed point, provided that M(x)^2 is Lipschitz continuous with a constant L.

4、Convergence Guarantee and Rate of Convergence
Claim: The iterative process exhibits exponential convergence under contraction mapping conditions.
Supporting Theory: Theorem 4.4 states that if T(x, y) is a contraction mapping, meaning that for two inputs x1 and x2, the update operator satisfies the inequality:
||T(x1, y) - T(x2, y)|| ≤ q * ||x1 - x2|| with q < 1,
then DEAL converges exponentially to a unique fixed point x-hat, with a decay rate proportional to q raised to the power of (k-1).

5、Stability with Respect to Input Perturbations
Claim: DEAL is stable under variations in the input measurements y, meaning that small perturbations in the input lead to bounded changes in the reconstruction.
Supporting Theory: Equation (18) states that for two different inputs y1 and y2, the difference between their corresponding reconstructions is bounded as follows:
||x-hat - z-hat|| ≤ (1 / (1 - q)) * (||H|| / lambda_epsilon) * ||y1 - y2||.
This result implies that DEAL produces consistent and stable reconstructions even when faced with measurement noise or variations in input data.

---

> ### Author Rebuttal · Authors · 2025-03-31
>
> We appreciate the reviewer’s feedback. We added more experiments and comparisons to highlight DEAL’s generalization and scalability (see response point 2 to reviewer dGc6 and response point [Questions For Authors] to reviewer Q6VC) and provided additional details on the attention mechanism (see responses point 2 and 4 to reviewer Q6VC). Our code repository will be released with the final paper.

---

### Official Review · Reviewer_dGc6 · 2025-03-23

**Overall Recommendation:** 4

**Summary:**

Summary

This paper proposes a novel Maximum a posteriori (MAP) method for solving linear inverse problems with Gaussian noise. The proposed method is based on Fields-of-Experts (FoE) regularization [1]. The authors suggest using quadratic potentials and learning the remaining parameters of the FoE regularization. The proposed algorithm is implemented as a neural network, and the regularization parameters are learned through end-to-end training.

Main findings:

- Using quadratic potential significantly accelerates the algorithm, enabling the effective learning of the regularization parameters.
- The regularization parameters learned from a denoising task can be directly applied to other linear inverse problems, such as super-resolution and MRI reconstruction, with minimal hyperparameter tuning.
- The proposed approach achieves reconstruction performance comparable to plug-and-play (PnP) methods.

[1] Roth, Stefan, and Michael J. Black. "Fields of experts: A framework for learning image priors." 2005 IEEE Computer Society Conference on Computer Vision and Pattern Recognition (CVPR'05). Vol. 2. IEEE, 2005.

## Update After Rebuttal

I want to thank the authors for their detailed response and for the effort they have put into addressing the concerns raised in the initial review. In light of the additional results and explanations provided by the authors, I have updated my score to **“4: Accept.”**

**Claims And Evidence:**

I think yes, the claims made in the submission supported by evidence.

**Essential References Not Discussed:**

I did not come across related works that are essential for understanding the key contributions of the paper but are not currently cited.

**Experimental Designs Or Analyses:**

Please refer to the Methods And Evaluation Criteria section for further details.

**Methods And Evaluation Criteria:**

I have some concerns regarding the evaluation.
1. The key novelty of the proposed method seems to be its ability to (almost) directly apply a model trained on denoising to other inverse problems. If this is the case, evaluating the model on denoising tasks may not fully capture its main contribution. It's useful to see how the proposed denoiser compares to state-of-the-art alternatives. However, these results could be presented as baseline comparisons rather than performance evaluations.
2. Since the proposed method requires fine-tuning of hyperparameters $\lambda$ and $\sigma$ for each task, comparing solely to plug-and-play methods may not provide a comprehensive perspective. I suggest including comparisons with recent state-of-the-art neural architectures specifically trained for super-resolution and MRI reconstruction. This would offer a more balanced view of the method's performance.
3. There appears to be a discrepancy in the reported PSNR values for DRUNet on the CBSD68 dataset. The reported PSNR values  (33.85 and 31.21 for CBSD68 with sigma 15 and 25 respectively) are approximately 0.45dB lower than those reported in the original DRUNet paper [1], which lists values of 34.30 and 31.69 under the same conditions. It would be helpful to clarify this inconsistency.
4. The reported results for DRIP in the color super-resolution experiment appear significantly lower than those presented in the original DRIP paper [1] (as shown in Table 8). Additional clarifications on the experimental setup and any differences in implementation would help understand this gap.

[1] Zhang, Kai, et al. "Plug-and-play image restoration with deep denoiser prior." IEEE Transactions on Pattern Analysis and Machine Intelligence 44.10 (2021): 6360-6376.

**Other Comments Or Suggestions:**

I have no further comments.

**Other Strengths And Weaknesses:**

Please refer to the Methods And Evaluation Criteria section for further details.

**Questions For Authors:**

I have no further questions for the authors.

**Relation To Broader Scientific Literature:**

The key contributions of the manuscript are related to the literature about PnP methods.

**Theoretical Claims:**

Yes

---

> ### Author Rebuttal · Authors · 2025-03-31
>
> We thank the reviewer for their time and valuable comments. To summarize our response:
> - We addressed the discrepancy of our numbers with the DPIR paper (due to cropping), we also added denoising results for the DPIR setup to underline the validity of our results.
> - We compare with a state-of-the-art neural network for super-resolution to offer a balanced view of DEAL's performance.
> - We adapt the writing of our paper to highlight the key points of the comments.
>
> If our responses have addressed your concerns, we would greatly appreciate it if you could reconsider your rating.
>
> > 1. The key novelty of the proposed method...
>
> Indeed, denoising is not the main performance benchmark, and we will adjust the writing to emphasize this more. In particular, our results for super-resolution and MRI are far more competitive and on par with SOTA *universal* (trained on denoising and applicable to other tasks) methods. Regarding novelty, beyond its easy extension to various tasks, DEAL bridges classic regularization with deep learning, offering interpretability, theoretical guarantees, and fewer parameters — all key advantages over other ML models. For this, our novel attention mechanism embedded into the FoE model is crucial; learning only the FoE components substantially deteriorates the performance (see response point 2 to reviewer Q6VC).
>
> > 2. Since the proposed method requires...
>
> Initially, we compared our method with SOTA PnP approaches, ensuring fair hyperparameter tuning across all evaluation tasks (see response point [Experimental Designs Or Analyses] of reviewer on6j). To showcase the benefits of universality, we adapt the super-resolution experiment to include an end-to-end trained transformer SwinIR [3], designed for a bicubic blur kernel. As expected, it is by far the best method within its training regime. However, its performance decreases substantially for other blur kernels and under the addition of noise. Likewise, for MRI, the literature shows that such models perform poorly outside of their training regime (e.g., DuDoRNet in Table 1 of [5]). Unfortunately, we could not find a well-performing end-to-end model for our specific setup (multi-coil, downsampling factor, noise). Generally, this lack of generalization for end-to-end methods is a major concern, and we highlight this with the new results in our paper.
>
> In the following table, #θ denotes the number of parameters in million, the triplets after PSNR are (s, $\sigma_n$, used kernels) where avg4 is the average of four different kernels. The two best numbers are bolded for each column.
>
> Method|Category|#θ|PSNR (2,0,bicubic)|PSNR (2,2.55,avg4)|PSNR (2,7.65,avg4)|PSNR (2,12.75,avg4)|PSNR (3,0,bicubic)|PSNR (3,2.55,avg4)|PSNR (3,7.65,avg4)|PSNR (3,12.75,avg4)|
> |-|-|-|-|-|-|-|-|-|-|-|
> |DEAL|Explicit Reg|**0.85**|**29.91**|**27.99**|**26.58**|25.75|**26.83**|**26.20**|**25.27**|24.59|
> |Prox-PnP|Conv. PnP|32.64|-|**27.93**|**26.61**|**25.79**|-|**26.13**|**25.29**|**24.67**|
> |IRCNN|PnP|**0.19**|29.84|26.97|25.86|25.45|26.74|25.60|25.72|24.38|
> |DPIR|PnP|32.64|29.63|27.79|**26.58**|**25.83**|26.70|26.05|**25.27**|**24.66**|
> |DiffPIR[4]|Diff. Model|93.56|29.73|27.84|26.48|25.63|-|-|-|-|
> |SwinIR[3]|End to End|11.94|**30.88**|24.56|22.84|20.73|**27.76**|22.41|21.24|19.53|
>
> > 3. There appears to be a discrepancy...
>
> Let us clarify the confusion. There are two different evaluation setups: we use 256x256 center-cropped images, as performed in ProxDrunet [2], while DPIR [1] uses full-size images. Our results match Table 1 of ProxDRUNet [2]. To cross-check the performance, we reran the evaluations within the setup of DPIR. Here, our results match exactly Table 2 of the DPIR paper [1]. Our chosen evaluation is detailed in Section 5.1, and we will revise the paper to avoid future confusion caused by the different setups.
>
> The two best numbers are bolded for each column.
>
> |Method|σₙ=5|σₙ=15|σₙ=25|
> |-|-|-|-|
> |BM3D|40.19|33.52|30.71|
> |DEAL(Ours)|40.33|**33.95**|**31.31**|
> |Prox-DRUNet|**40.40**|33.91|31.14|
> |DNCNN|-|33.90|31.24|
> |DRUNet|**40.59**|**34.30**|**31.69**|
>
> > 4. The reported results for DPIR...
>
> We use the setup of [2] and our results are following their Table 4. There are three differences with Table 8 of DPIR: (i) we report the metrics on 256x256 center-cropped CBSD68 images; (ii) the noise levels are different; (iii) we report the average over the 4 kernels (a-d). The configuration is described in Section 5.2, and we will adapt the writing to prevent future confusion caused by the different setups.
>
> Refs.
>
> [1] Zhang et al. "PnP image restoration with deep denoiser prior.", 2021.
>
> [2] Hurault et al. "Proximal denoiser for convergent plug-and-play optimization with nonconvex regularization.", 2022.
>
> [3] Liang et al. "Swinir: Image restoration using swin transformer.", 2021.
>
> [4] Zhu et al. "Denoising diffusion models for PnP image restoration.", 2023.
>
> [5] Song et al. "Solving inverse problems in medical imaging with score-based generative models, 2021.

---

> > ### Comment · Reviewer_dGc6 · 2025-04-04
> >
> > As highlighted in the authors' rebuttal, SwinIR trained for bicubic blur performs significantly better than DEAL when the inference task settings match those used during training. However, its performance declines when inference conditions, such as the blur kernel or added noise, differ from the training setup. It suggests that SwinIR requires retraining or adaptation to maintain optimal reconstruction performance across different settings.
> >
> > I am curious whether DEAL can be applied consistently across all super-resolution tasks using the same hyperparameters or if, like SwinIR, it requires parameter tuning for different scenarios.
> >
> > Additionally, regarding DEAL's key advantages over other ML methods. DEAL solves a full-size optimization problem for each input sample during inference. How does this impact the computation complexity of the inference? Specifically, how does the runtime scale as input image size increases?

---

> > > ### Author Response · Authors · 2025-04-05
> > >
> > > Dear Reviewer,
> > >
> > > Thank you for your response.
> > >
> > > > I am curious whether DEAL can be applied consistently across all super-resolution tasks using the same hyperparameters or if, like SwinIR, it requires parameter tuning for different scenarios.
> > >
> > >
> > > In DEAL's evaluation setup, we only need to adjust the **scalar** hyperparameter $\lambda$ to the data noise level $\sigma_n$. Then, we use the same $\lambda$ for all blur kernels and downsampling ratios. We follow the same principle for other universal methods when fine-tuning their hyperparameters (in accordance with their papers and implementations). We compare against two SwinIR models from their codebase, which are trained in a noiseless setup for rates s=2, s=3, separately. No weights for the noisy super-resolution tasks are provided and SwinIR does not offer interpretable hyperparameters for noise adaptation. Retraining the models requires significantly more data and computational resources compared to the minimal hyperparameter fine-tuning of the universal approaches.
> > >
> > > To provide further evidence for the good generalization of DEAL to new kernels, we propose the following experiment: A noise-less superresolution task with a downsampling rate of s=2, where the SwinIR is trained on the bicubic kernel and DEAL's $\lambda$ is fine-tuned on the bicubic task. Then, we apply SwinIR and DEAL to new kernels (A-D, different Gaussian kernels) with no further change for both models. Here, we explicitly see that SwinIR needs retraining for new kernels while DEAL still performs well with the hyperparameters tuned on the bicubic kernel. The following table contains the reconstruction PSNR for the noiseless super-resolution task with s=2 for different blur kernels on centered-cropped CBSD68 data.
> > > |Method|Kernel Bicubic|Kernel A|Kernel B|Kernel C|Kernel D|
> > > |-|-|-|-|-|-|
> > > |DEAL|29.91|29.59|29.76|28.57|27.21|
> > > |SwinIR|30.88|25.72|25.85|24.50|23.66|
> > >
> > > Regarding the adaptation of DEAL's $\lambda$ to the noise level $\sigma_n$, we can also use a theoretically motivated closed-form formula for $\lambda$. The performance drop is marginal compared to the fine-tuned case. The following table presents the average reconstruction PSNR over 4 kernels for the super-resolution task with s=2 on centered-cropped CBSD68 data.
> > > |Method|$\sigma_n=2.55$|$\sigma_n=7.65$|$\sigma_n=12.75$
> > > |-|-|-|-
> > > |DEAL (fine-tuned)|27.99|26.58|25.75
> > > |DEAL ($\lambda = 0.1 + 0.035\sigma_n^2$)|27.97|26.57|25.75
> > >
> > > Additionally, one can retrain DEAL in an end-to-end manner by incorporating the forward operator, which will close the significant gap in the training regime of SwinIR. However, this is outside the scope of our paper which tries to position DEAL among universal approaches. Furthermore, adapting end-to-end models to different tasks (e.g., MRI and super-resolution) often necessitates architectural changes—this is not the case for DEAL or other universal approaches.
> > >
> > > > Additionally, regarding DEAL's key advantages over other ML methods. DEAL solves a full-size optimization problem for each input sample during inference. How does this impact the computation complexity of the inference? Specifically, how does the runtime scale as input image size increases?
> > >
> > > A similar question regarding the scalability of DEAL was raised by Reviewer Q6VC. To address this, we did experiments on images ranging from size 256x256 to 2048x2048, reporting memory and time usage in our rebuttal (please see the response [Questions for Authors] in the rebuttal to Reviewer Q6VC). Our results demonstrate that DEAL consistently outperforms Prox-DRUNet, the most comparable method in terms of universality, performance, and convergence properties. In particular, the optimization process for DEAL is efficient: (i) we rely on CG which is known as an optimal solver given the structure of our $A_k$; (ii) each update in the refinement process is warm-started with the previous solution; therefore, the later CGs in the pipeline need very few iterations to converge.
> > >
> > > Finally, we would like to emphasize the **interpretability** of DEAL. In Section 6 of our paper (Figures 7 and 8), we provide two interpretations of DEAL and its backbone attention mechanism. Figure 8 is particularly striking, as it demonstrates the exact relationship between the input measurements and DEAL's output. Specifically, each pixel of this output is a just a weighted average of the measurements, where the weights are the products of DEAL's refinement. These weights adapt well to the structure of the image. We believe that this interpretability is an elegant qualitative feature of DEAL, which goes beyond numerical metrics and opens the door to more well-performing, explainable image reconstruction methods.
> > >
> > > We sincerely appreciate your consideration and look forward to your final evaluation.

---

### Decision · Program_Chairs · 2025-05-01

**Decision:**

Accept (poster)

**Comment:**

The paper presents a well-supported approach to image reconstruction, combining classic regularization with deep learning. After reviewing both the critiques and the rebuttal, I believe the concerns raised by the critical reviewer have been adequately addressed. The method shows strong theoretical grounding, solid performance, and practical benefits like robustness and interpretability. I’m recommending a weak accept, but ask the authors to revise the final version to clarify comparisons with task-specific baselines, address any inconsistencies in reported results, and provide more detail on computational cost and scalability.